# Match or Replay: Self Imitating Proximal Policy Otimization

## Abstract

Reinforcement Learning (RL) agents often struggle with inefficient exploration, particularly in environments with sparse rewards. Traditional exploration strategies can lead to slow learning and suboptimal performance because agents fail to systematically build on previously successful experiences, resulting in poor sample efficiency. To tackle this issue, we propose a self-imitating on-policy algorithm aimed at enhancing exploration and sample efficiency by leveraging past high-reward state-action pairs to guide policy updates. Our method incorporates self-imitation by utilizing optimal transport distance in dense reward environments to prioritize state visitation distributions matching the most rewarding trajectory. For sparse reward environments, we uniformly replay successful self-encountered trajectories to facilitate structured exploration. Experimental results across diverse environments demonstrate substantial improvements in learning efficiency, including MuJoCo for dense rewards and the partially observable 3D Animal-AI Olympics and multi-goal Point-Maze for sparse rewards. Our approach achieves faster convergence and significantly higher success rates compared to state-of-the-art self-imitating RL baselines. These findings underscore the potential of self-imitation as a robust strategy for enhancing exploration in RL, with applicability to more complex tasks.

## 1 Introduction

Deep Reinforcement Learning (DRL) (Li, 2017) has achieved remarkable success in solving complex problems across a variety of domains, including robotic manipulation (Han et al., 2023), flight control (Kaufmann et al., 2018), intelligent perception system (Chaudhary et al., 2023), and real-time strategy game-play (Andersen et al., 2018). However, despite these advancements, DRL algorithms still face significant challenges in efficient learning, resulting in poor sample inefficiency (Baker et al., 2019). A major contributing factor is the reliance on unguided exploration to discover optimal policies, leading to slow convergence and excessive policy divergence, particularly in environments with sparse rewards.

Guided exploration using expert demonstrations has been proposed as a potential solution. Approaches such as those in (Salimans & Chen, 2018; Ecoffet et al., 2019; Xu et al., 2023; Duan et al., 2017; Zhou et al., 2019; Haldar et al., 2023) have explored the use of expert data to guide the agent's learning process. However, these methods often suffer from challenges such as acquiring expert demonstrations, the risk of bias, and the potential for suboptimal policies when demonstrations are not sufficiently informative.

This work addresses these challenges by exploiting the agent's past successful state-action transitions to guide exploration. To this end, we propose a novel Self-Imitation Learning (SIL) (Oh et al., 2018) approach that leverages the agent's past high-reward transitions to guide current policy updates—effectively "bootstrapping" its learning process. This strategy enhances exploration and reduces the risk of divergence from effective behaviors (Lin, 1992). By leveraging the most rewarding past trajectories, our approach prevents the agent from deviating too far from previously learned successful behaviors, improving exploration and learning efficiency.

Self-imitation learning has been applied to diverse complex tasks such as robotics (Luo & Schomaker, 2023; Luo et al., 2021), text-based games (Shi et al., 2023), procedurally generated environments (Lin et al., 2023), interactive navigation (Kim et al., 2023), and large language models (Xiao et al., 2024). Despite

these advancements, a unified on-policy RL approach that effectively integrates self-imitation across both state- and pixel-based observations while accommodating dense and sparse reward settings remains absent. Prior works such as (Oh et al., 2018) have explored self-imitation in reinforcement learning; however, they primarily focus on off-policy algorithms or incorporate off-policy elements when adapting self-imitation to on-policy methods like PPO (Schulman et al., 2017). In contrast, we introduce Self-Imitating Proximal Policy Optimization (SIPP)—a framework explicitly designed for on-policy RL, seamlessly integrating self-imitation within PPO's update mechanism without relying on replay buffers or off-policy corrections. By doing so, SIPP preserves PPO's stability and theoretical guarantees while significantly enhancing exploration and sample efficiency.

For dense reward environments, we propose the Match strategy. RL agents can struggle with sample efficiency due to high-dimensional state and action spaces, leading to slow convergence or suboptimal policies even with dense rewards. As implemented in SIPP, self-imitation addresses this by guiding the agent toward high-value regions of the state space, reducing policy divergence, and accelerating learning. Specifically, the Match strategy uses optimal transport (Peyré et al., 2019), particularly the sinkhorn algorithm (Cuturi, 2013), to measure the similarity between state distributions of the current policy and the most rewarding episodic rollout from the past. By prioritizing state-action transitions that closely match these distributions, the Match strategy ensures that exploration focuses on the self-encountered regions of the state space with high expected rewards.

For sparse or binary reward environments, we introduce the Replay strategy, which maintains an imitation buffer storing self-encountered successful trajectories. Instead of relying completely on the agent's current behavior, the replay strategy prioritizes past high-reward trajectories by adding them strategically to the learning process. This approach views self-imitation as an advantage-based prioritization, where the agent focuses on proven successful experiences. By emphasizing high-reward trajectories over arbitrary past transitions, the Replay strategy ensures that the agent benefits from useful past data while maintaining the stability of the on-policy learning framework. This selective sampling allows the agent to improve its policy without the risk of destabilizing updates, even in the absence of dense reward signals.

To summarize, our key contributions are as follows:

- Self-imitating on-policy algorithm: We propose Self-Imitating Proximal Policy Optimization (SIPP), a novel self-imitation learning algorithm that enhances exploration and sample efficiency in dense and sparse reward settings.

- Optimal transport-based prioritization: We introduce the Match strategy, which uses Optimal Transport (Peyré et al., 2019) and the Sinkhorn algorithm (Cuturi, 2013) to prioritize state-action transitions that closely match the state distribution of the most rewarding past episodic rollout, improving learning efficiency in dense reward environments.

- Replay strategy for sparse rewards: We develop a Replay strategy that stores and reuses successful trajectories to reinforce long-term dependencies and improve learning from delayed rewards, resulting in enhanced sample efficiency.

- Mitigation of policy divergence: Our approach mitigates excessive policy divergence by bootstrapping RL policy learning with past successful behaviors, stabilizing and enhancing the learning process.

- Diverse empirical validation: We validate SIPP through experiments across a wide range of environments, including complex MuJoCo (Towers et al., 2023) tasks, multi-goal PointMaze navigation (de Lazcano et al., 2023), and partially observable 3D Animal-AI Olympics (Crosby et al., 2019), demonstrating significant improvements in learning efficiency and performance over existing methods (Oh et al., 2018; Gangwani et al., 2018).

## 2 Related Work

Many attempts have addressed the sample efficiency and exploration problem in reinforcement learning. However, this literature has divided the long work history mainly into guided and unguided exploration.

**Guided exploration** paradigms aim to exploit expert trajectories to address RL agents' sample efficiency and exploration problems. Recently, in this direction, (Sontakke et al., 2024) presented an approach that uses a single demonstration and distilled knowledge contained in Video-and-Language Models (VLMs) to train a robotics policy. They use VLMs to generate rewards by comparing expert trajectories and policy rollouts. Another single demonstration guided approach was presented by (Libardi et al., 2021) for solving three-dimensional stochastic exploration. They exploit expert trajectories and value-estimate prioritized trajectories to learn optimal policy under uncertainty. Similarly, (Salimans & Chen, 2018) trained a robust policy using a single demonstration by replaying the demonstration for $n$ steps, after which agents learned in a self-supervised manner. To make the agent robust to randomness, they monotonically decrease the replay steps $n$. (Uchendu et al., 2023) presents an expert-guided learning. They employ two policies to solve tasks: the guide policy and the exploration policy. The guide policy introduces a curriculum of initial states for the exploration policy, significantly easing the exploration challenge and facilitating rapid learning. As the exploration policy becomes more proficient, the reliance on the guide policy diminishes, allowing the RL policy to develop independently and continue improving autonomously. This progressive reduction in guide-policy influence enables the agent to transition to a fully autonomous exploration phase, enhancing its long-term performance and adaptability.

(Xu et al., 2023) uses expert demonstration to improve exploration in learning from demonstrations in sparse reward settings. They assign an exploration score to each demonstration, generate an episode, and train a policy to imitate exploration behaviors. (Nair et al., 2018) design an auxiliary objective on demonstrations to solve hard exploration problems and anneal away the demonstration guidance once the policy performs better than the demonstration. (Huang et al., 2023) used a two-component approach: a novel actor-critic-based policy learning module that efficiently uses demonstration data to guide RL exploration and a non-parametric module that employs nearest-neighbor matching and locally weighted regression for robust guidance propagation at states distant from the demonstrated ones.

**Unguided exploration** approaches use self-experience, count-based methods, or some type of prioritized experience replay buffer to guide policy in hard exploration problems. In this literature, we focus only on approaches under the paradigm of Self-Imitation Learning (SIL) coined by (Oh et al., 2018). (Oh et al., 2018) presented an approach for self-imitation learning for off-policy algorithms. They store experiences in a replay buffer and learn to imitate state-action pairs in the replay buffer only when the return in the past episode is greater than the agent's value estimate. They also extended their approach to the on-policy algorithm. However, the proposed algorithm does not have a strong theoretical connection to the on-policy algorithms.

(Gangwani et al., 2018) introduces the Stein Variational Policy Gradient (SVPG), a self-imitating algorithm designed for on-policy reinforcement learning. In this approach, policy optimization is framed as a divergence minimization problem, where the objective is to minimize the difference between the visitation distribution of the current policy and the distribution induced by experience replay trajectories with high returns. The method incorporates an auxiliary objective that regularizes this divergence, allowing for improved exploration and more effective policy updates. However, their experiments are limited to episodic, delayed, or noisy reward settings, which may restrict the generalizability of their results to more complex environments.

(Chen & Lin, 2020) presents a SIL technique for off-policy algorithms. In their approach, they provide a constant reward at each step in addition to an episodic environment reward. Further, they maintain two replay buffers, one with $K$ highest episodic reward trajectories and the other with all agent-generated trajectories, and sample from these two replay buffers to train the policy. They limit their work to delayed episodic rewards. (Tang, 2020) presents a self-imitation learning approach for off-policy learning by extending the traditional Q-learning with a generalized n-step lower bound. They adopt SIL by leveraging trajectories where the behavior policy performs better than the current policy. (Ferret et al., 2020) proposes a self-imitating variant of DQN for dense reward environments. In this approach, they propose to adopt self-imitation using a modified reward function. They augment the true reward with a weighted advantage term, the difference between a true discounted reward and an expected future return.

(Kang & Chen, 2020) introduces the Explore-then-Exploit (EE) framework, which integrates Random Network Distillation (RND) (Burda et al., 2018) and Generative Adversarial Self-Imitation Learning (GASIL) (Guo et al., 2018). The framework tackles the exploration-exploitation trade-off by using RND to facilitate

exploration and prevent the policy from stagnating at local optima. At the same time, GASIL accelerates policy convergence by leveraging past successful trajectories. Rather than directly combining these methods, which could confuse the agent, the authors propose an interleaving approach, where the agent switches between exploration and imitation based on specific criteria.

Recently, (Li et al., 2023) extended the SIL approach to Goal-Conditioned Reinforcement Learning. They achieve this by designing a target action values function that can effectively combine the training mechanism of self-initiated policy and actor policy. SILP (Luo et al., 2021) method utilizes a planning mechanism for robotic manipulation that identifies good policies from previous experiences, allowing the agent to imitate high-quality actions even when explicit demonstrations are unavailable. By incorporating planning into the SIL framework, the agent can efficiently explore and exploit past successful behaviors. The approach improves the exploration-exploitation balance and enhances learning stability.

The proposed approach aligns with unguided exploration with a focus on on-policy learning. The proposed approach uses past experiences to bootstrap policy learning, making a strong connection with the self-imitation learning paradigm.

## 3   Preliminaries

**Markov Decision Process** We formulate the problem as a Markov Decision Process (MDP) $M$ : $\langle S, A, T, R, \gamma, p(s_0) \rangle$ with $S = \{s_1, ..., s_n\}$ being set of environment states, $A = \{a_1, ..., a_n\}$ being set of agents actions, $T : S \times A \to S$ is state transition function, $R : S \times A \times S \to \mathbb{R}$ is reward function, $\gamma \in (0, 1)$ is discount factor, and $p(s_0)$ is environments initial state distribution. At each time step $t$, the agent observes the state of the environment $s_t$, samples an action $a_t$ from the action set $A$, and receives a reward $r_t$ from the environment. The reinforcement learning agent is trained to maximize the expected reward collected from agent-environment interaction (Sutton & Barto, 2018).

## 4   Method

In this work, we propose an approach that can guide an agent's exploration by combining the agent's current behavior and past successful trajectories for an on-policy RL algorithm. Unlike prior self-imitation methods that rely on off-policy data or modify the reward function, SIPP operates entirely within the on-policy framework. By modifying the rollout buffer sampling strategy (Match) or selectively replaying successful trajectories (Replay), SIPP reuses successful trajectories in a controlled fashion and integrates them into PPO's on-policy training loop, without requiring additional target networks or density-ratio corrections typical in off-policy RL. While this introduces some bias from sample reuse, as discussed in works such as GePPO (Queeney et al., 2021), empirical results indicate that it remains stable and effective. This seamless integration with PPO ensures that our approach is both stable and efficient. The key highlights are as follows:

- Our approach does not alter the base RL policy nor introduce new separate models requiring training, unlike prior works Gangwani et al. (2018); Kang & Chen (2020).

- Our approach does not modify the true reward, preventing any bias in learning, unlike (Chen & Lin, 2020).

- We address exploration by self-imitation for dense, sparse, and binary rewards encompassing state and pixel-based observations, unlike (Oh et al., 2018; Gangwani et al., 2018), which addressed only delayed and noisy rewards for state-based observation in an on-policy setting.

### 4.1   *Match*: Self-Imitating Proximal Policy

The *Match* strategy addresses the exploration for dense reward environments. The proposed approach connects with self-imitation learning by prioritizing transitions (state-action) similar to the past most rewarding episodic rollout. We find the similarity or match between trajectories using optimal transport.

---

**Algorithm 1** *Match*: Self-Imitating Proximal Policy

---

Initialize parameters $\theta$
Initialize $\mathcal{B}_\mathcal{I} \leftarrow \{\}$
Initialize rollout buffer $\mathcal{D} \leftarrow \{\}$

1: **for** every update **do**
2:     Collect rollouts via policy $\pi$
3:     Update $\mathcal{B}_\mathcal{I}$ by dynamically storing the single best trajectory (highest return) seen so far.
4:     Compute advantage estimates $\hat{A}_1, \hat{A}_2, ..., \hat{A}_T$ for trajectories in $\mathcal{D}$
5:     **for** every epoch **do**
6:         Sample transitions from $\mathcal{D}$ uniformly or weighted by Equation 3, controlled by IET ($\epsilon$)
7:         Optimize $L^{PPO}$ w.r.t. $\theta$: $\theta \leftarrow \theta - \eta \nabla_\theta L^{PPO}$; where $\eta$ is learning rate
8:     **end for**
9:     Empty rollout buffer $\mathcal{D} \leftarrow \{\}$
10: **end for**

---

Optimal Transport (Cuturi, 2013; Peyré & Cuturi, 2020; Luo et al., 2023) offers a structured approach for comparing probability distributions. The squared Wasserstein distance between two discrete distributions, $\nu_x = \frac{1}{T} \sum_{t=1}^{T} \delta_{x_t}$ and $\nu_y = \frac{1}{T'} \sum_{t'=1}^{T'} \delta_{y_{t'}}$, is given by:

$$W^2(\nu_x, \nu_y) = \min_{\nu \in \zeta} \sum_{t=1}^{T} \sum_{t'=1}^{T'} c(x_t, y_{t'}) \nu_{t,t'}, \tag{1}$$

where $\zeta = \left\{ \nu \in \mathbb{R}^{T \times T'} : \nu \mathbf{1} = \frac{1}{T} \mathbf{1}, \nu^T \mathbf{1} = \frac{1}{T'} \mathbf{1} \right\}$ represents the set of coupling matrices, $c$ is a cost function, and $\delta_x$ is the Dirac measure for $x$. The optimal coupling $\nu^*$ effectively matches the samples from $\nu_x$ and $\nu_y$. Unlike measures such as the KL-divergence (Kullback & Leibler, 1951), the Wasserstein distance is a metric that considers the space's geometry.

Consider $\hat{q}_e = \frac{1}{T'} \sum_{t'=1}^{T'} \delta_{s_t^e}$ and $\hat{q}_\pi = \frac{1}{T} \sum_{t=1}^{T} \delta_{s_t^\pi}$ which represent the empirical state distributions of an expert policy $\pi_e$ and a behavior policy $\pi$, respectively. The squared Wasserstein distance between the expert and behavior policies is then given by:

$$W^2(\hat{q}_\pi, \hat{q}_e) = \min_{\nu \in \zeta} \sum_{t=1}^{T} \sum_{t'=1}^{T'} c(s_t^\pi, s_{t'}^e) \nu_{t,t'} \tag{2}$$

Let $\nu^*$ denote the optimal coupling for this problem. The distance between each state can then be calculated as:

$$d_{OT}(s_t^\pi) = -\sum_{t'=1}^{T'} c(s_t^\pi, s_{t'}^e) \nu_{t,t'}^*, \tag{3}$$

The proposed approach does not rely explicitly on expert policy $\pi_e$. Instead, we adopt a more flexible framework by considering the distribution of the best episodic rollout generated by the behavior policy $\pi$ as the surrogate expert policy. This removes the dependency on external expert trajectories and leverages the agent's high-performing experiences. We achieve this by maintaining an imitation buffer $\mathcal{B}_\mathcal{I}$, which dynamically stores the most rewarding episodic rollout encountered by the behavior policy. The state visitation distribution of the trajectory stored in $\mathcal{B}_\mathcal{I}$ serves as the expert distribution. This design ensures that the imitation buffer evolves continuously as the agent discovers better-performing trajectories, thus adapting the surrogate expert distribution over time.

Building on this, we compute the Wasserstein distance (Equation 3) between the state visitation distribution in the imitation buffer $\mathcal{B}_\mathcal{I}$ and the current state visitation distribution generated by the behavior policy.

---

**Algorithm 2** *Replay*: Self-Imitating Proximal Policy

---

Initialize parameters $\theta$
Initialize $\mathcal{B}_{\mathcal{I}} \leftarrow \{\}$
Initialize rollout buffer $\mathcal{D} \leftarrow \{\}$

1: **for** every update **do**
2:     Sample $\tau$ from $\{\mathcal{B}_{\mathcal{I}}, \text{Env}\}$ controlled by IET $(\epsilon)$
3:     **if** $\tau \in \mathcal{B}_{\mathcal{I}}$ **then**
4:         Sample a trajectory from $\mathcal{B}_{\mathcal{I}}$ and store transition in $\mathcal{D}$
5:     **else if** $\tau \in \text{Env}$ **then**
6:         Execute current policy and store transition in $\mathcal{D}$
7:     **end if**
8:     Compute advantage estimates $\hat{A}_1, \hat{A}_2, ..., \hat{A}_T$ for trajectories in $\mathcal{D}$
9:     Optimize $L^{PPO}$ wrt $\theta$ for $K$ epoch $\theta \leftarrow \theta - \eta \nabla_\theta L^{PPO}$; where $\eta$ is learning rate
10:    Update $\mathcal{B}_{\mathcal{I}}$
11:    Empty rollout buffer $\mathcal{D} \leftarrow \{\}$
12: **end for**

---

This distance measures alignment, quantifying how closely the agent's trajectory matches its past successful experiences. Using this metric, each transition in the rollout buffer $\mathcal{D}$ is assigned a sampling priority proportional to the computed distance. Transitions that are more similar to those in $\mathcal{B}_{\mathcal{I}}$ are given higher priority, effectively guiding the agent toward replicating its best-performing behaviors.

The effective balance between exploration (sampling diverse transitions) and exploitation (focusing on high-priority transitions) is governed by the hyperparameter Imitation-Exploration Trade-off (IET) coefficient $\epsilon$. IET coefficient controls the probability of sampling transition for policy updates from rollout buffer $\mathcal{D}$ either uniformly or with a probability governed by Equation 3. This prioritization framework enhances sample efficiency and ensures that the agent's learning process is bootstrapped by past successful experiences, eliminating the need for external supervision. Algorithm 1 provides a detailed overview of the optimal transport-based trajectory-matching strategy.

## 4.2 *Replay*: Self-Imitating Proximal Policy

This section addresses the exploration challenge for sparse reward settings using self-imitation. Sparse reward environments often present significant challenges for learning agents due to the limited availability of positive feedback. To mitigate this challenge, we propose the *Replay* strategy, a variant of the *Match* strategy, tailored specifically for sparse reward scenarios. Unlike *Match*, which uses past trajectories to generate preferences for current trajectories, the *Replay* strategy focuses on directly replaying successful past behaviors.

While prior self-imitation methods prioritize individual state-action pairs, *Replay* introduces a trajectory-level replay mechanism tailored for sparse reward settings. By replaying entire successful trajectories rather than isolated experiences, adopted from (Libardi et al., 2021), *Replay* improves learning in sparse-reward settings by repeatedly exposing the agent to high-return trajectories, which effectively reinforces key state-action dependencies over time. This complements the agent's ability to learn long-horizon behavior. Additionally, as demonstrated in the Animal-AI Olympics experiments, by design *Replay* can handle partial observability, making it more versatile than existing methods.

In this context, "replay" refers to including trajectories stored in the imitation buffer $\mathcal{B}_{\mathcal{I}}$ into the rollout buffer $\mathcal{D}$ and treating them as potential behaviors generated under the current policy. This ensures that the agent strategically encounters successful trajectories at each policy update to guide policy learning, even in sparse reward environments where most interactions yield little to no reward.

*Replay* strategy maintains an imitation buffer $\mathcal{B}_{\mathcal{I}}$ to store past successful trajectories. The imitation buffer is initialized as empty, and as the agent interacts with the environment, trajectories are added to $\mathcal{B}_{\mathcal{I}}$ based on the return for sparse reward and the FIFO strategy for binary reward environments, respectively. Each trajectory in $\mathcal{B}_{\mathcal{I}}$ is represented as $\tau = (s_0, a_0, s_1, a_1, ...)$, where policy $\pi_{\mathcal{I}}$ associated with the imitation buffer

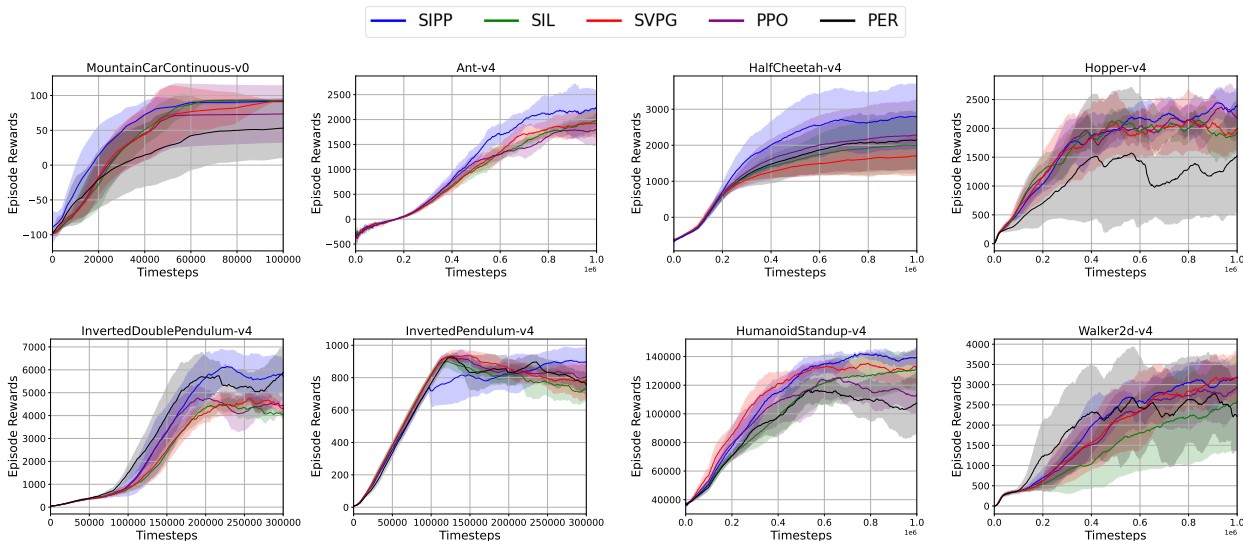

Figure 1: Results show the performance of 8 MuJoCo (Towers et al., 2023) continuous control tasks (refer to Figure 7 for results on all tasks). The plots are the learning curves and show the episodic rewards along the y-axis, evaluated through the current policy. The reported results are the mean across seven different seeds with shaded regions highlighting standard deviation. The proposed algorithms outperform all baselines across all tasks by being competitive or better than others.

maps observations from the imitation buffer to the corresponding actions in the stored successful trajectories with a probability of one while assigning zero probability to all other actions. The policy $\pi_{\mathcal{I}}$ is used to replay trajectories from $\mathcal{B}_{\mathcal{I}}$, integrating them into the rollout buffer $\mathcal{D}$.

At each policy update epoch, trajectories are sampled from $\mathcal{B}_{\mathcal{I}}$ with probability $\epsilon$, or generated by the current policy with probability $1 - \epsilon$. These trajectories are then added to the rollout buffer $\mathcal{D}$ from which transitions are sampled to update the policy. This trajectory sampling mechanism ensures a balance between exploration and exploitation. The IET coefficient $\epsilon$ plays a crucial role in tuning this balance. A higher value of $\epsilon$ emphasizes exploitation by prioritizing trajectory sampling from $\mathcal{B}_{\mathcal{I}}$, while a lower value encourages exploration by focusing on trajectories generated from the agent's most recent interactions.

The presented *Replay* strategy, hence, offers a robust framework for addressing sparse and binary reward challenges by ensuring that successful behaviors are continually reinforced in policy learning. This selective reuse of high-value trajectories enhances sample efficiency and mitigates the risk of the agent getting stuck in unproductive exploration loops.

With this explicit focus on replaying successful past behaviors, the *Replay* strategy aligns with the principles of self-imitation learning while addressing the unique challenges posed by sparse reward environments. Algorithm 2 provides a detailed outline of implementing this strategy, highlighting its integration into the policy optimization process.

## 5 Experiments

In this section, we aim to answer the following questions:

- Does bootstrapping policy learning, with its few past experiences, enhance sample efficiency and hard exploration across diverse tasks?

- Can a single past successful behavior be sufficient to guide policy learning in complex sequential continuous control tasks?

- Is replaying past successful trajectories sufficient for policy learning in multi-goal and partially observable, sparse reward settings?

## 5.1 Implementation Details

For dense reward environments in MuJoCo (Towers et al., 2023), we implement the *Match* strategy. The network architecture utilizes a multi-layer perceptron (MLP) with two hidden layers containing 64 units and *tanh* as the activation function. The PPO policy is updated over 10 epochs per training iteration. Training batches are sampled uniformly or with priority based on the optimal transport distance between the current trajectories and past best episodic rollouts, controlled by the Imitation-Exploration Trade-off coefficient $\epsilon$ during each epoch. The imitation buffer is initiated with size 1. Further details about the hyperparameters and implementation can be found in the supplementary material Table 2.

For sparse reward settings, such as the multi-goal PointMaze (de Lazcano et al., 2023) navigation tasks, we adopt the *Replay* strategy. This setup shares the same MLP-based architecture used in MuJoCo environments. Similarly, for the Animal-AI Olympics (Crosby et al., 2019), a partially observable 3D environment with binary rewards, we apply the *Replay* strategy but with a different architecture: a three-layer convolutional neural network (CNN). The input comprises the last four stacked frames ($84 \times 84$ RGB pixels). In both PointMaze and Animal-AI tasks, the rollout buffer is populated with trajectories sampled either from the current policy with probability $1 - \epsilon$ or from the imitation buffer $\mathcal{B}_\mathcal{I}$ with probability $\epsilon$. The imitation buffer is initialized with size 10. Comprehensive details of the hyperparameters for all environments are provided in the supplementary material Table 3.

## 5.2 Choice of Baselines:

Self-imitation learning (SIL) has been mainly explored to enhance exploration in off-policy reinforcement learning (RL) algorithms with a limited focus on on-policy RL algorithms. Further, recent works Luo & Schomaker (2023); Xiao et al. (2024); Shi et al. (2023) predominantly focus on problem-specific adoption of SIL rather than advancing SIL from a broader algorithmic perspective. Notably, there remains a limited analysis of SIL's potential in the context of on-policy RL algorithms to solve diverse problem settings, leaving a significant gap in understanding its general applicability. This limits the choice of baseline in the context of our problem.

**PPO:** PPO is vanilla proximal policy algorithm (Schulman et al., 2017), which does not impose any self-imitation learning paradigm.

**SIL-PPO:** SIL (Oh et al., 2018) is an off-policy RL algorithm that imitates past state-action pairs that have higher returns than agent value estimates. The proposed approach was also extended to the on-policy PPO (Schulman et al., 2017) algorithm with a focus on dense or delayed rewards. Further, as highlighted by (Oh et al., 2018), SIL lacks a theoretical connection with on-policy algorithms.

**SVPG-PPO:** SVPG (Gangwani et al., 2018) is a self-imitating on-policy algorithm that uses Stein variational gradient descent to minimize the divergence between the current policy's visitation distribution and that of past high-return trajectories. Unlike SIPP, SVPG introduces an auxiliary objective that regularizes this divergence, which can complicate the learning process.

**PER-PPO**: Prioritize Experience Replay (PER) (Schaul et al., 2015) technique uses TD-error based transition prioritization. We extended this method to PPO, where we prioritized samples in the rollout buffer based on TD-error. We use a strategy similar to our method to balance exploration and exploitation.

## 5.3 Performance of *Match* on Continuous Control Tasks

In this section, we investigate the effect of self-imitation on continuous control tasks with dense rewards. We evaluate the performance of our *SIPP-Match* strategy on 10 MuJoCo (Towers et al., 2023) tasks with chosen baselines.

Compared with all the baselines, the performance of the *Match* strategy on continuous control tasks is shown in Figure 1. The proposed *Match* algorithm outperforms PPO (Schulman et al., 2017) and SIL-PPO across all

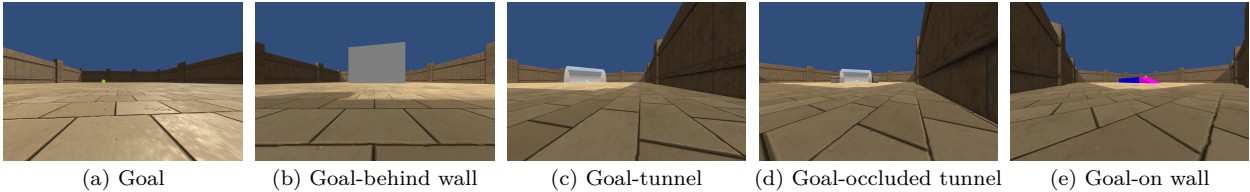

(a) Goal        (b) Goal-behind wall        (c) Goal-tunnel        (d) Goal-occluded tunnel        (e) Goal-on wall

Figure 2: All tasks feature one goal and one agent. The positions of the agent and goal are selected randomly at the start of each episode from a predefined set of fixed initial positions. Each episode initializes the environment by sampling these positions, ensuring variability while maintaining a structured distribution. There is only one source of reward per environment, i.e., a binary reward is provided for reaching the goal. The agent observes the arena through a first-person view with partial visibility, reflecting the limitations of a partially observable environment.

the tasks, with SVPG+PPO lagging in most tasks except having competitive performance on Walker2d-v4 and Humanoid-v4 tasks. PER (Schaul et al., 2015) uses a TD-error-based strategy, which uses value function estimates to prioritize transitions and lag across all the tasks.

In MuJoCo benchmark environments, the agent benefits from continuous feedback via a smooth and dense reward structure, facilitating faster exploration and learning. Despite this, our experiments demonstrate that the optimal transport distance prioritized self-bootstrapping can further enhance exploration for the proximal policy. By prioritizing the most informative experiences, our method ensures that the agent focuses on high-value learning opportunities, accelerating convergence and improving policy robustness.

Additionally, the proposed approach stores only the states visited by the most rewarding past episodic rollout, making it straightforward compared to prior methods. Unlike our approach, (Oh et al., 2018) compares returns of past experiences with agent value estimates to select experiences for self-imitation, which can be noisy and introduce bias in policy learning (Libardi et al., 2021; Raileanu & Fergus, 2021). Furthermore, (Gangwani et al., 2018) uses Stein variational gradient descent as a regularizer to minimize divergence between state-action visitation distributions of the current policy and past rewarding experiences. However, their approach introduces bias in policy learning, which they address by simultaneously learning multiple diverse policies.

In summary, the proposed *Match* algorithm integrates seamlessly with PPO without introducing additional learning parameters and requires only one trajectory to guide self-imitation learning. It introduces a single hyperparameter, IET coefficient ($\epsilon$), to control whether training batches are uniformly sampled or prioritized using optimal transport distance based on past successful episodic rollouts. The single hyperparameter provides a simple mechanism to balance exploration and exploitation. This approach offers a practical and efficient solution to enhance reinforcement learning performance in complex environments.

### 5.4    Performance of *Replay* in Sparse Reward Tasks

In this section, we empirically evaluate self-imitation performance in sparse and binary reward settings. We believe that self-imitation can play a crucial role in such reward settings, as the ability of an agent to reach some success can be extremely difficult with sparse rewards. The previous works were limited to MuJoCo environments with dense or delayed rewards. Motivated by this, we evaluate the performance of *SIPP-Replay* on a diverse set of tasks, including multi-goal gymnasium-robotics PointMaze navigation sparse reward environments (de Lazcano et al., 2023) and partially observable 3-dimensional Animal-AI Olympics (Crosby et al., 2019) binary reward environments.

### 5.4.1    Task Definitions:

The PointMaze environment is a 2-dimensional maze. We use two variants of the PointMaze environment. First, with fixed agent position and varying goal position, i.e., the goal position is reinitialized at each episode. Second, both the goal and agent positions are reinitialized after every reset.

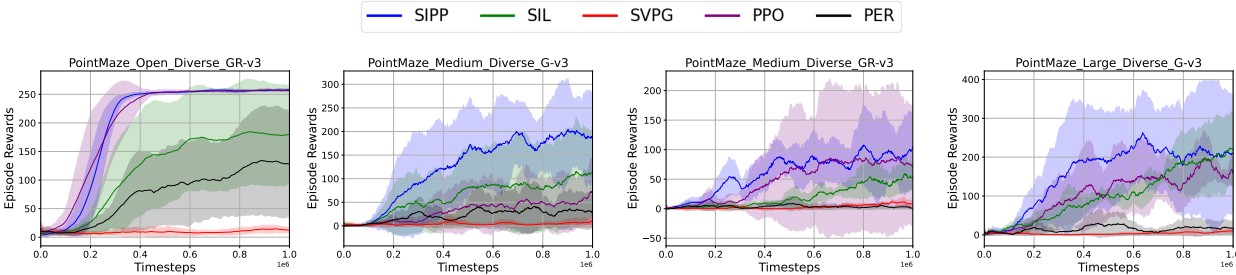

Figure 3: Results show the performance on 4 PointMaze (de Lazcano et al., 2023) multi-goal sparse reward tasks (refer to Figure 8 for results on all tasks). The plots are the learning curves and show the episodic rewards along the y-axis, evaluated through the current policy. The reported results are the mean across seven different seeds. The proposed algorithms outperform all the baselines by a significant margin.

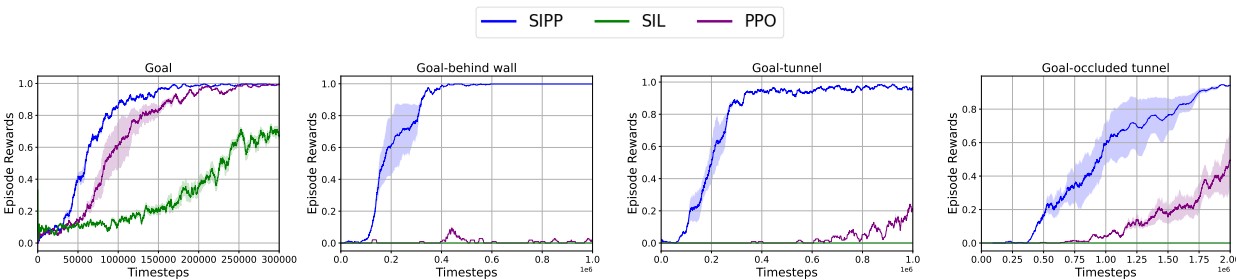

Figure 4: Results show the performance on the 4 Animal-AI Olympics environment (Crosby et al., 2019) binary reward tasks (refer to Figure 9 for results on all tasks). The plots are the learning curves and show the episodic rewards (success rate) along the y-axis, evaluated through the current policy. The reported results are the mean across 5 seeds, with shaded regions highlighting the standard deviation. The proposed algorithms outperform PPO by a significant margin.

The Animal-AI Olympics (Crosby et al., 2019) is a partially observable 3-dimensional environment where an agent can navigate freely inside an arena. We designed a total of 5 experiments. Each experiment has a different level of complexity based on the type of obstacles present in the arena. The descriptions of playgrounds are as follows :

- **Goal:** In this arena, the agent has to reach the goal position. The agent and goal can be anywhere in the arena. There are no obstacles in the arena.

- **Goal-behind wall**: The goal is hidden behind a wall in this arena. The agent and goal positions are different in each configuration. The agent needs to learn to find the goal, which is hidden behind the wall.

- **Goal-tunnel**: This arena has a transparent tunnel that is open from both ends. The agent can not penetrate the tunnel walls and must enter the tunnel to find the goal.

- **Goal-occluded tunnel:** This arena is identical to the previous one, except that the tunnel entrances are occluded with movable boxes. The agent must learn to move the boxes to find the goal inside the tunnel.

- **Goal-on wall:** In this arena, we place the goal on an L-shaped wall. The agent must learn to find a ramp to climb up the wall and avoid falling off the wall to reach the goal.

### 5.4.2 Empirical Analysis:

The performance of *SIPP-Replay* is shown in Figures 3 and 4. The choice of baselines for the PointMaze environment is consistent with the MuJoCo tasks, as both involve fully observable MDPs. However, for the

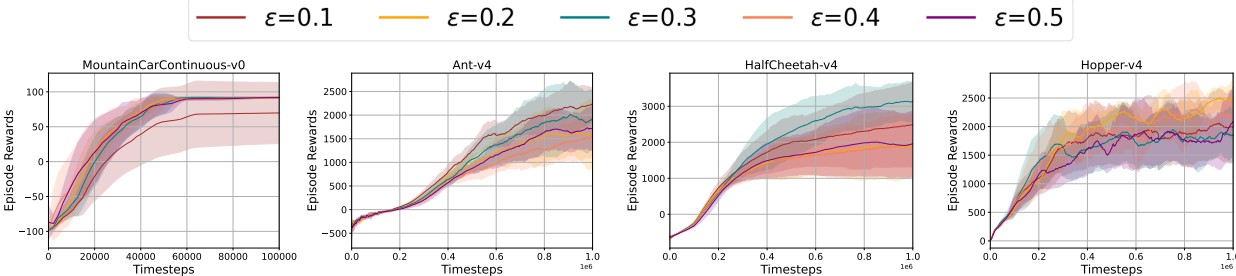

Figure 5: Results show the ablation study on 4 MuJoCo (Towers et al., 2023) continuous control tasks (refer to Figure 10 for complete results). The parameter $\epsilon$ controls the degree of exploration vs exploitation. The plots are the learning curves and show the episodic rewards along the y-axis evaluated through the current policy with different $\epsilon$. The reported results are the mean across 5 seeds, with shaded regions highlighting the standard deviation.

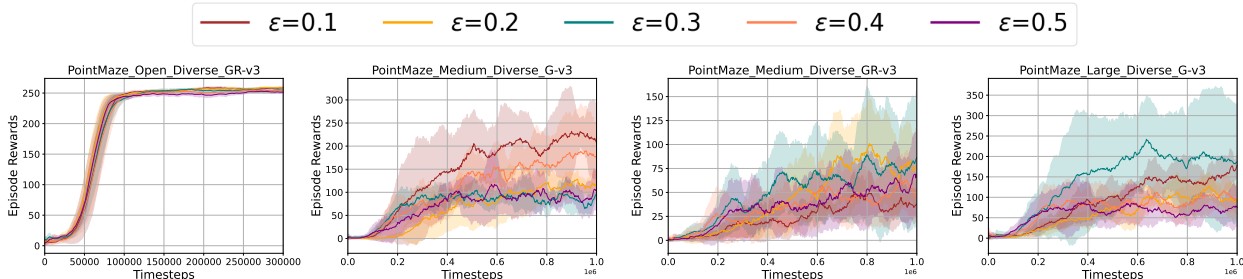

Figure 6: Results show the ablation study on 4 PointMaze (de Lazcano et al., 2023) multi-goal sparse reward tasks (refer to Figure 11 for complete results). The parameter $\epsilon$ controls the replay frequency to balance exploration vs exploitation. The plots are the learning curves and show the episodic rewards along the y-axis evaluated through the current policy with different $\epsilon$. The reported results are across 5 different seeds.

Animal-AI Olympics task, the choice of baselines is restricted to PPO. This limitation arises because the official implementations of baselines, SIL and SVPG, are tailored specifically to fully observable environments like MuJoCo and do not extend support to the Animal-AI Olympics.

Further, Baselines such as SIL and SVPG rely on explicit divergence estimation or advantage comparisons over fully observed states, making them less suitable for partially observable environments. In contrast, SIPP's replay strategy treats stored trajectories as expert demonstrations and integrates them directly into PPO's training without requiring density estimation or changes to the underlying architecture. This allows SIPP to operate naturally in POMDP settings, where modeling state visitation distributions is non-trivial or infeasible.

To ensure fairness, we adapted the SIL baseline (Oh et al., 2018) by using observation histories as proxies for states in the Animal-AI Olympics environment. However, even with this adaptation, SIL performed worse than vanilla PPO. This degradation stems from SIL's reliance on either dense reward signals or access to true states to guide its imitation strategy—conditions that are not satisfied in our partially observable setting. This comparison highlights the broader applicability of our method: SIPP does not assume full observability or reward density and is, to our knowledge, the first self-imitation approach successfully deployed in such a complex, diverse POMDP environment.

In our experiments, the Imitation-Exploration Trade-off coefficient IET ($\epsilon$) was set to 0.3 for all PointMaze tasks except for PointMaze_Medium_Diverse_G, where $\epsilon = 0.1$ was used based on preliminary experiments. The performance of our proposed algorithm exceeds all the baselines on the Maze navigation task, as shown in Figure 3. We believe that the credit assignment problem plays a crucial role under sparse reward conditions,

and we tackle this by replaying past trajectories. Unlike baseline methods, which prioritize state-action pairs, SIPP focuses on episodic trajectory-level prioritization rather than state visitation distributions. This helps the agent to understand which actions contribute to future rewards. To incorporate successful trajectories in the replay buffer, we consider it as the possible behavior of the agent in the current environment. The *Replay* strategy results in even superior performance, Figure 4, for partially observable environments due to its inherent design capability to adapt to partial observability. The results show that PPO agents encounter some success but fail to learn due to policy instability. Policy instability refers to the divergence of PPO's policy from successful behaviors due to frequent updates with new data, which overwrite past successes. However, the proposed *Replay* strategy stores those behaviors and keeps replaying them. This reinforces those successful behaviors in policy learning, and the agent eventually learns to mimic those successful behaviors.

To summarize, it is evident from the results that self-imitation can help agents learn in both dense and sparse reward settings. In a dense reward setting, prioritizing state visitation that matches the past successful state is sufficient, as a dense reward structure can guide the agent to learn the long-term consequences of actions taken in those states. However, a more exploitative strategy is required in sparse reward settings, such as replaying past successful episodic trajectories.

### 5.5 Tuning Self-Imitation vs. Exploration

The proposed strategy uses the exploitation of the agent's past behaviors. However, it is crucial to explore for the agent to learn an optimal policy. This balance between exploitation and exploration in SIPP is achieved through the Imitation-Exploration Trade-off (IET) coefficient $\epsilon$. In *SIPP-Match* this parameter indicates the probability of sampling training batches uniformly or with a priority proportional to the optimal transport distance with the most successful past trajectory. In *SIPP-Replay* this parameter controls the trajectory replay probability from the imitation buffer $\mathcal{B}_\mathcal{I}$.

The effect of IET for *SIPP-Match* strategy is shown in Figure 5. The ablation study shows maximum performance improvement for $\epsilon = 0.1, 0.2$ or $0.3$ across all tasks. This highlights the importance of exploration as a more greedy imitation results in sub-optimal performance. However, for simpler tasks such as MountainCar-Continuous, the performance is mostly similar as these simpler environments require less exploration, and even an aggressive imitation strategy results in similar performance.

A similar analysis was performed to find the balance between the exploration and exploitation trade-off of the PointMaze navigation environments. The results of this ablation study are shown in Figure 6. We didn't perform a similar analysis of the Animal-AI Olympics environment. However, the IET coefficient for the Animal-AI Olympics environment was kept $\epsilon = 0.3$ based on the insights from the above ablation studies.

In conclusion, Our ablation study on $\epsilon$ highlights its influence on SIPP's performance. In dense reward environments like MuJoCo, smaller $\epsilon = 0.1, 0.2$ perform well, as frequent rewards naturally guide exploration. Conversely, in sparse reward settings like PointMaze, a higher value $\epsilon = 0.3$ improves outcomes by emphasizing imitation of rare successful trajectories. These results indicate that $\epsilon$ should be adjusted based on the task's reward structure and exploration demands.

## 6 Conclusion

This paper proposed a self-imitating proximal policy framework to address exploration and credit assignment challenges in dense and sparse reward environments. Through extensive experimentation, we demonstrated that bootstrapping policy learning with past rewarding experiences effectively mitigates forgetting behavior and reduces policy divergence, leading to enhanced exploration and stability. The simplicity and efficacy of the proposed algorithm highlight its versatility across different problem settings. Furthermore, we showed that self-imitation and exploration are inherently complementary, enabling agents to leverage prior successes for guided learning, which can be crucial in hard exploration tasks.

**Limitation and Future Direction:** Our results suggest that dynamically integrating self-imitation with adaptive exploration strategies can provide a promising avenue for future research. Extending the proposed

framework to more complex tasks, including partially observable, multi-agent, and hard-exploration settings, could unlock further potential and establish broader applicability across reinforcement learning domains. However, the potential of the proposed approach can be limited due to the risk of getting trapped in local minima. We believe these limitations can be addressed by integrating self-imitation with hard exploration or intrinsic reward module strategies.

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

# A  Integrating SIPP with Hard Exploration Techniques

In this section, we investigate the integration of the Self-Imitating Proximal Policy (SIPP) with Random Network Distillation (RND), a technique designed to enhance exploration in reinforcement learning by encouraging agents to visit novel states. This combination leverages SIPP's self-imitation mechanism, which reinforces past successful behaviors, and RND's intrinsic motivation, which promotes exploration, to improve performance in environments with challenging exploration requirements.

Our approach combines two core components:

- Self-Imitating Proximal Policy (SIPP): SIPP enhances policy optimization by prioritizing high-return trajectories from the agent's past experiences. In sparse reward settings, such as Atari games, we employ the Replay strategy, maintaining an imitation buffer of successful trajectories (defined by cumulative extrinsic rewards) that are selectively replayed during policy updates using Proximal Policy Optimization (PPO).

- Random Network Distillation (RND): RND generates an intrinsic reward signal based on the prediction error between a fixed random network and a trainable network, incentivizing the agent to explore novel states by quantifying their unfamiliarity.

In the RND+SIPP framework, the agent's total reward at each timestep is the sum of the extrinsic reward from the environment and the intrinsic reward from RND. The policy is updated via PPO with the SIPP Replay strategy, where the imitation buffer stores trajectories based solely on their cumulative extrinsic rewards. This ensures that SIPP reinforces behaviors leading to tangible environmental success while RND independently drives the exploration of novel regions, mitigating potential conflicts between exploitation and exploration objectives.

We evaluated RND+SIPP on three Atari 2600 games from the Arcade Learning Environment—Gravitar, Venture, and Solaris—selected for their sparse rewards and exploration challenges. The results of RND were taken from (Burda et al., 2018). We did not perform an ablation study on the SIPP hyperparameter for this study and kept the IET coefficient fixed across tasks to 0.1 and the imitation buffer size 1. In the Gravitar task, RND+SIPP achieves an 11.7% improvement, leveraging SIPP's reinforcement of successful trajectories alongside RND's exploration. Venture Performance is comparable, with a slight 2.5% decrease, suggesting task-specific tuning of $\epsilon$ may be needed. In Solaris, A 9.3% gain highlights the benefit of combining imitation with exploration in complex state spaces.

Table 1: Performance Comparison on hard exploration tasks.

| Task | RND | RND+SIPP |
|---|---|---|
| Gravitar | 3906 | **4363** |
| Venture | **1859** | 1813 |
| Solaris | 3282 | **3589** |

The integration of SIPP with RND demonstrates that combining self-imitation learning with intrinsic motivation provides a dual benefit. On the one hand, SIPP ensures that the agent leverages its past successes to stabilize policy updates. On the other hand, RND continually drives the agent to explore unvisited or less familiar regions of the state space. The trade-off between these components is controlled via the Imitation-Exploration Trade-off coefficient, allowing for task-specific tuning.

The analysis suggests that while RND alone can foster exploration, it may not prevent the divergence of effective strategies. RND+SIPP overcomes this limitation by continually reinforcing high-value behaviors,

leading to improved overall performance. Future work may involve dynamically adapting the balance between intrinsic and extrinsic rewards based on the observed learning dynamics, thereby further refining the exploration-exploitation trade-off.

# B    Preliminaries and Background

This section provides a detailed overview of the key components that form the basis of our Self-Imitating Proximal Policy (SIPP) algorithm. We discuss Proximal Policy Optimization (PPO), Optimal Transport theory (with emphasis on the Wasserstein metric), advantage estimation, and the role of value networks. This background sets the stage for understanding the technical innovations of our approach.

## B.1    Proximal Policy Optimization (PPO)

Proximal Policy Optimization (PPO) Schulman et al. (2017) is a first-order policy gradient method designed to improve the stability of on-policy learning. The core idea is to restrict the policy update so that the new policy does not deviate too much from the old policy. PPO achieves this by optimizing the clipped surrogate objective:

$$L^{\text{CLIP}}(\theta) = \mathbb{E}_t \left[ \min \left( r_t(\theta) A_t, \ \text{clip} \left( r_t(\theta), 1 - \zeta, 1 + \zeta \right) A_t \right) \right], \tag{4}$$

where $r_t(\theta) = \frac{\pi_\theta(a_t|s_t)}{\pi_{\theta_{\text{old}}}(a_t|s_t)}$ is the probability ratio, $A_t$ is the advantage estimate at time $t$, and $\zeta$ is a hyperparameter that controls the extent of the policy update. This clipping mechanism prevents excessively large updates, thereby balancing exploration and exploitation while ensuring sample efficiency.

## B.2    Optimal Transport and the Wasserstein Distance

Optimal Transport (OT) is a mathematical framework for comparing probability distributions by finding the minimal "cost" of transporting one distribution to match another Cuturi (2013); Peyré et al. (2019). In our work, we utilize the Wasserstein distance, a metric derived from OT, to measure the similarity between state visitation distributions. Given two discrete distributions $\nu_x = \frac{1}{T} \sum_{t=1}^T \delta_{x_t}$ and $\nu_y = \frac{1}{T'} \sum_{t'=1}^{T'} \delta_{y_{t'}}$ the squared 2-Wasserstein distance is defined as:

$$W^2(\nu_x, \nu_y) = \min_{\nu \in \zeta} \sum_{t=1}^T \sum_{t'=1}^{T'} c(x_t, y_{t'}) \nu_{t,t'}, \tag{5}$$

where $c(x_t, y_t') = \|x_t - y_t'\|^2$ is the cost function (typically the squared Euclidean distance), and $\zeta(\nu_x, \nu_y)$ denotes the set of all couplings with marginals $\nu_x$ and $\nu_y$. In practice, the Sinkhorn algorithm is often employed to compute an entropic regularized version of this distance efficiently. In our algorithm, the Wasserstein distance is used to quantify the mismatch between the current policy's state visitation and that of the best past trajectories, thereby guiding self-imitation.

## B.3    Advantage Estimation

The advantage function is central to policy gradient methods as it quantifies the relative benefit of taking an action in a given state compared to the average performance. Formally, the advantage function is defined as:

$$A(s, a) = Q(s, a) - V(s), \tag{6}$$

where $Q(s, a)$ is the action-value function representing the expected return when taking action $a$ in state $s$, and $V(s)$ is the state-value function estimating the expected return from state $s$ under the current policy. Accurate estimation of $A(s, a)$ is critical for effective learning. In practice, techniques such as Generalized Advantage Estimation (GAE) Schulman et al. (2015) are employed to balance bias and variance, providing robust advantage estimates that drive policy improvement.

### B.4 Value Networks

Value networks are neural networks used to approximate the state-value function $V(s)$, which predicts the expected cumulative reward from a given state. In actor-critic architectures, the value network plays two primary roles: (i) it serves as a baseline to reduce the variance of the policy gradient estimates, and (ii) it provides feedback for policy evaluation. The value network is typically trained by minimizing the squared error between its predictions and the empirical returns:

$$L_{\text{value}}(\theta_v) = \mathbb{E}_t \left[ (V_{\theta_v}(s_t) - R_t)^2 \right], \tag{7}$$

where $R_t$ denotes the observed return and $\theta_v$ represents the parameters of the value network. A well-calibrated value network is essential for reliable advantage estimation and overall training stability.

**Integration in SIPP:** In our Self-Imitating Proximal Policy (SIPP) algorithm, these components are tightly integrated. PPO forms the backbone of our policy updates, while Optimal Transport guides the prioritization of experiences based on the similarity of state distributions. The advantage function and value networks are instrumental in quantifying and propagating high-value behaviors. Together, these elements enable our framework to effectively leverage past successful experiences to improve exploration and sample efficiency.

## C  Guidelines for Tuning $\epsilon$-Greedy Parameter in SIPP

The hyperparameter $\epsilon$ governs the balance between imitation and exploration in SIPP. To tune it effectively, we suggest the following:

- **Baseline Setting:** Start with $\epsilon = 0.3$, which works reasonably well across diverse tasks.

- **Dense Reward Tasks:** For environments with frequent rewards (e.g., MuJoCo), reduce $\epsilon$ to 0.1–0.2 to prioritize refining known behaviors over excessive exploration.

- **Sparse Reward Tasks:** In hard exploration scenarios (e.g., PointMaze), increase $\epsilon$ to 0.3–0.5 to leverage imitation of scarce successes.

- **Stability Check:** If training shows a high variance in success rates, consider reducing $\epsilon$ to stabilize learning via stronger imitation.

- **Cross-Validation:** For optimal results, perform a grid search over values like 0.1, 0.3, and 0.5, especially in critical applications.

These guidelines, grounded in our empirical findings, should enhance SIPP's reproducibility and usability.

## D  Ethical Considerations and Broader Impacts

As with any reinforcement learning (RL) technology, the deployment of our Self-Imitating Proximal Policy (SIPP) algorithm requires careful consideration of both ethical concerns and broader societal impacts. Below, we briefly outline these issues in alignment with current best practices in RL research.

**Safety and Robustness:** RL algorithms can exhibit unpredictable behavior in dynamic, real-world environments, potentially leading to unsafe outcomes. Although SIPP leverages past successful trajectories to guide policy updates and mitigate risky exploration, conducting extensive validation across diverse scenarios remains essential. Rigorous testing and robust safety verification protocols should be integrated before deployment, especially in safety-critical applications such as autonomous systems and robotics.

**Fairness and Bias:** Algorithms that learn from historical data risk perpetuating existing biases. In resource allocation or automated decision-making contexts, biased training trajectories could reinforce discriminatory

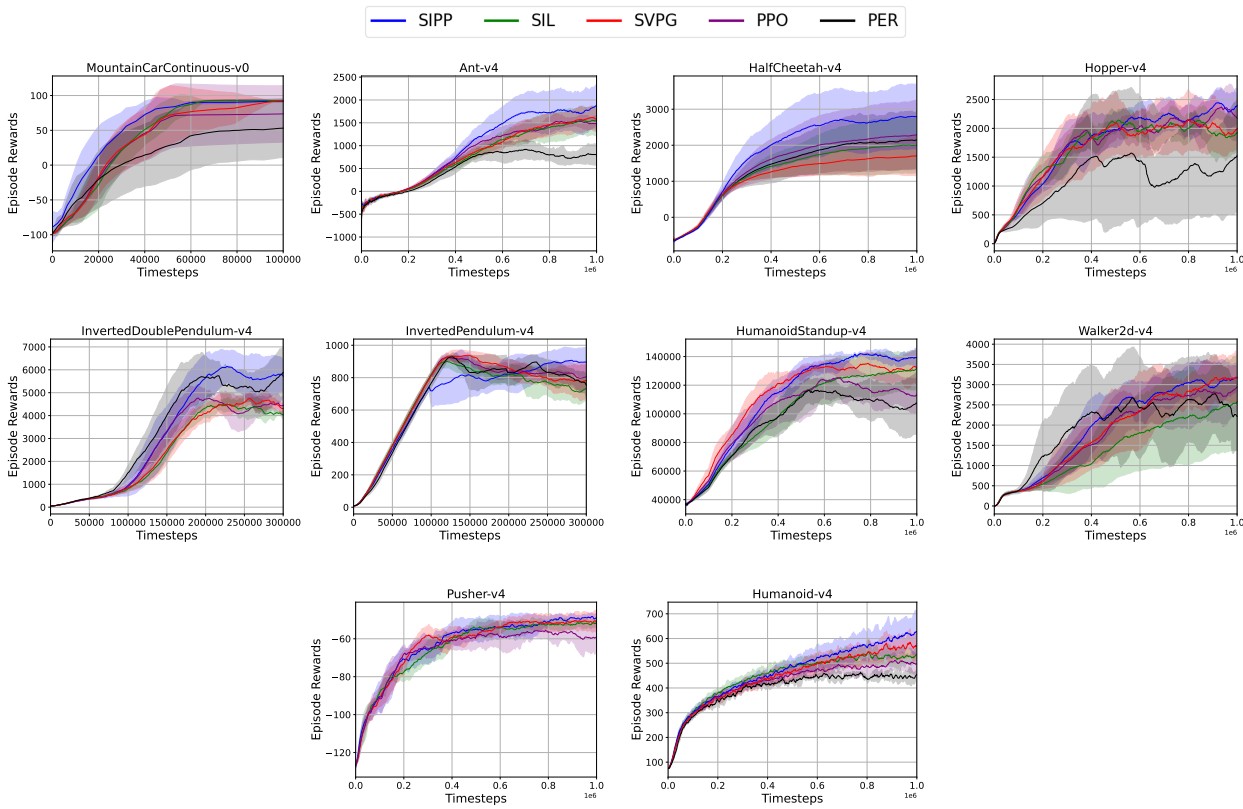

Figure 7: Results show the performance of 10 MuJoCo (Towers et al., 2023) continuous control tasks. The plots are the learning curves and show the episodic rewards along the y-axis, evaluated through the current policy. The reported results are the mean across seven different seeds. The proposed algorithms outperform all baselines across all tasks by being competitive or better than others.

patterns. To address this, it is important to implement fairness-aware reward functions and regularly audit model behavior, ensuring that the system does not inadvertently favor certain groups or outcomes.

**Privacy and Data Security:** When RL methods are applied in sensitive domains (e.g., healthcare, finance), the use of stored trajectories raises privacy concerns. Data anonymization techniques, differential privacy, and secure data handling practices are crucial to protect individual information while still benefiting from self-imitation learning. Moreover, ensuring the integrity of stored data against adversarial manipulation is vital for maintaining system reliability.

**Environmental Impact:** The computational demands of training deep RL models can contribute to a significant environmental footprint. Researchers should consider energy-efficient practices and the potential trade-offs between model complexity and resource consumption. Exploring more efficient algorithmic designs or leveraging renewable energy sources for computation may help mitigate these impacts.

**Beneficial Applications:** Despite the challenges, SIPP holds promise for enhancing sample efficiency and improving policy robustness in a range of applications, from robotics and disaster response to healthcare and scientific research. By fostering more reliable and effective RL systems, our work aims to contribute positively to these fields, provided that ethical considerations are integrated throughout the development and deployment lifecycle.

In summary, while our approach offers substantial benefits, its ethical deployment requires ongoing collaboration among researchers, practitioners, and policymakers to ensure that advancements in RL are both socially responsible and aligned with broader public interests.

Table 2: Parameters For Animal-AI Olympics Environment

| Parameter | Values |
| --- | --- |
| episode length | 1000 |
| image size (RGB) | $84 \times 84 \times 3$ |
| initial reward threshold | 0 |
| frame-skip | 2 |
| frame-stack | 4 |
| discount factor ($\lambda$) | 0.99 |
| gae-gamma ($\gamma$) | 0.95 |
| value loss coefficient ($c_1$) | 0.1 |
| entropy loss coefficient ($c_2$) | 0.02 |
| learning rate | $10^{-4}$ |
| ppo-epoch | 4 |
| number-mini-batch | 7 |
| value-clip | 0.15 |
| policy-clip | 0.15 |
| buffer size ($\mathcal{B_I}$) | 10 |

Table 3: PPO Hyper-parameters

| Parameter | Values |
| --- | --- |
| learning rate | $3e^{-4}$ |
| n-steps | 2048 |
| batch size | 64 |
| n-epochs | 10 |
| discount factor | 0.99 |
| gae-gamma | 0.95 |
| clip-range | 0.2 |
| normalize advantage | True |
| vf-coef | 0.5 |
| max-grad-norm | 0.5 |

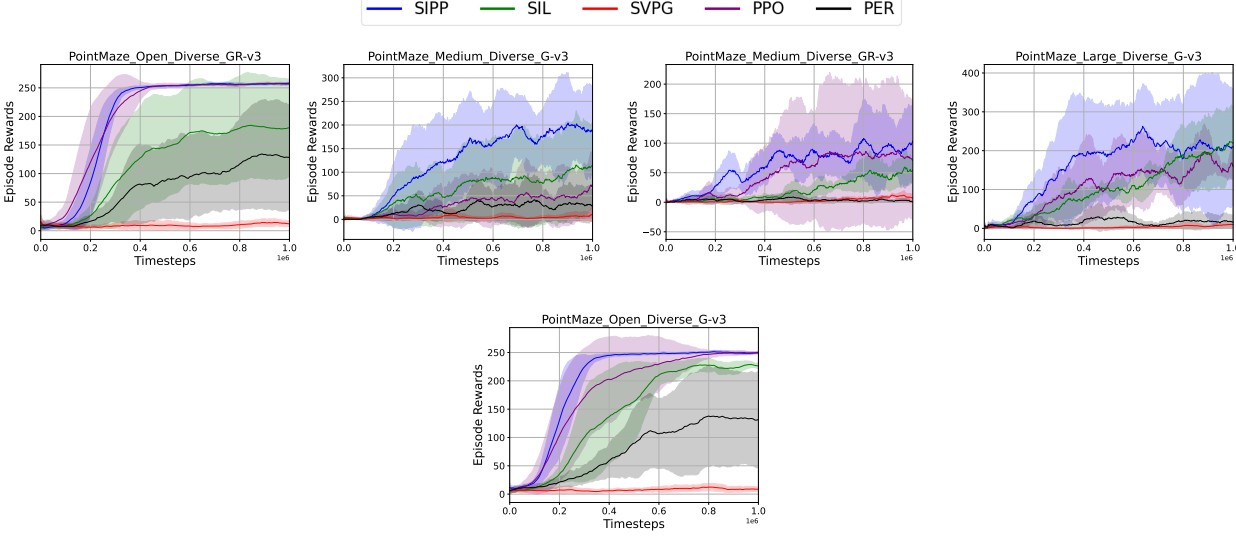

Figure 8: Results show the performance on all 5 PointMaze multi-goal sparse reward tasks. The plots are the learning curves and show the episodic rewards along the y-axis, evaluated through the current policy. The reported results are across seven different seeds. The proposed algorithms outperform all the baselines by a significant margin.

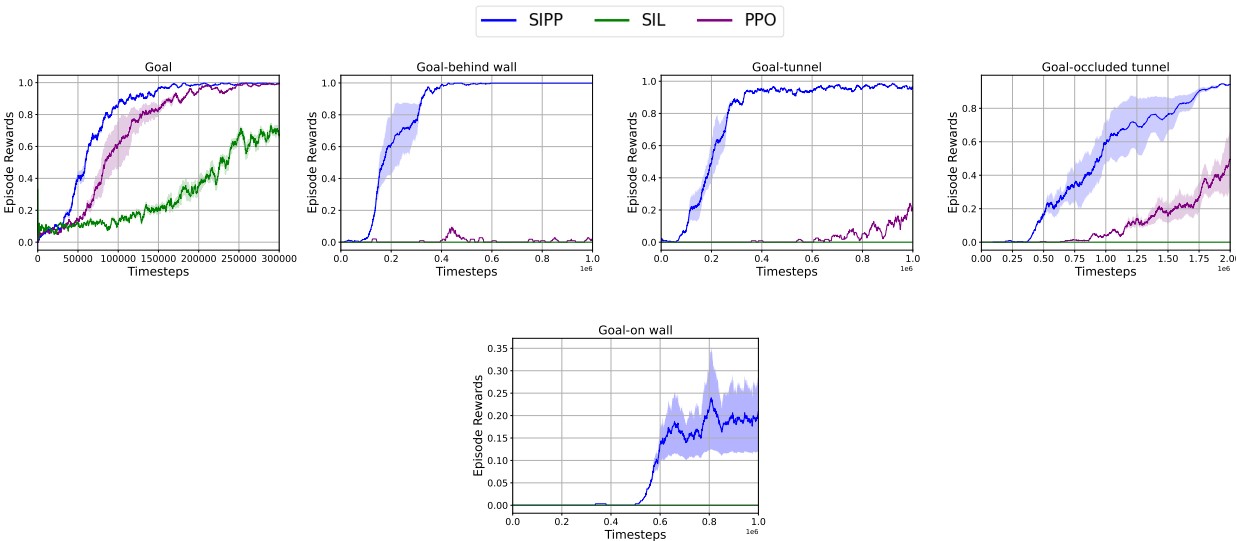

Figure 9: Results show the performance on all 5 Animal-AI Olympics environments sparse reward tasks. The plots are the learning curves and show the episodic rewards (success rate) along the y-axis, evaluated through the current policy. The reported results are across 5 different seeds. The proposed algorithms outperform all the baselines by a significant margin.

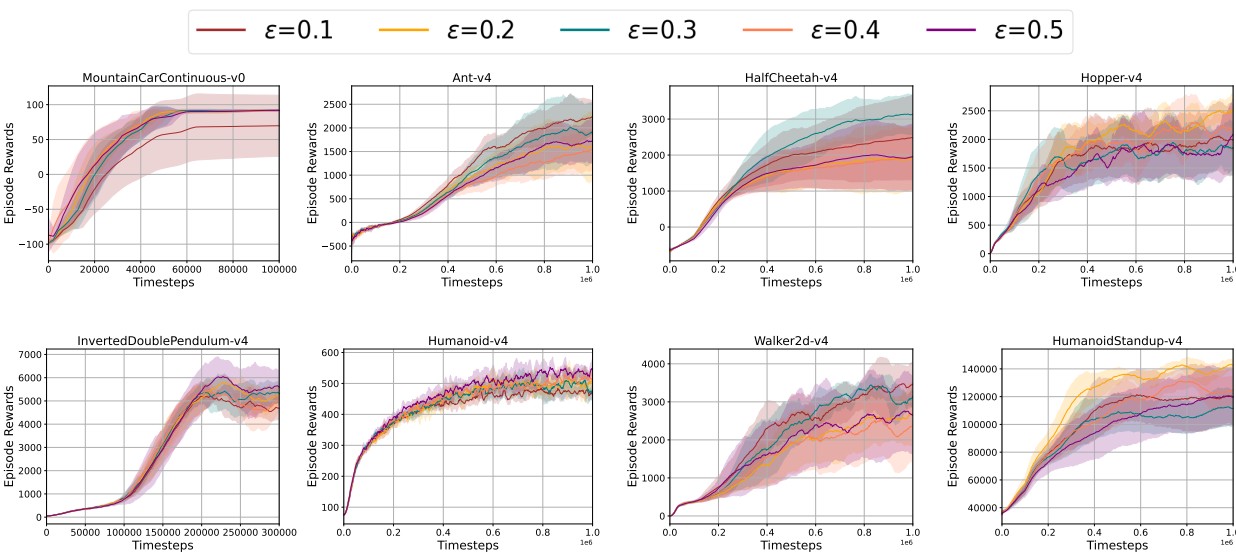

Figure 10: Results show the ablation study on 8 MuJoCo (Towers et al., 2023) continuous control tasks. The parameter $\epsilon$ controls the degree of exploration vs exploitation. The plots are the learning curves and show the episodic rewards along the y-axis evaluated through the current policy with different $\epsilon$. The reported results are the mean across five different seeds.

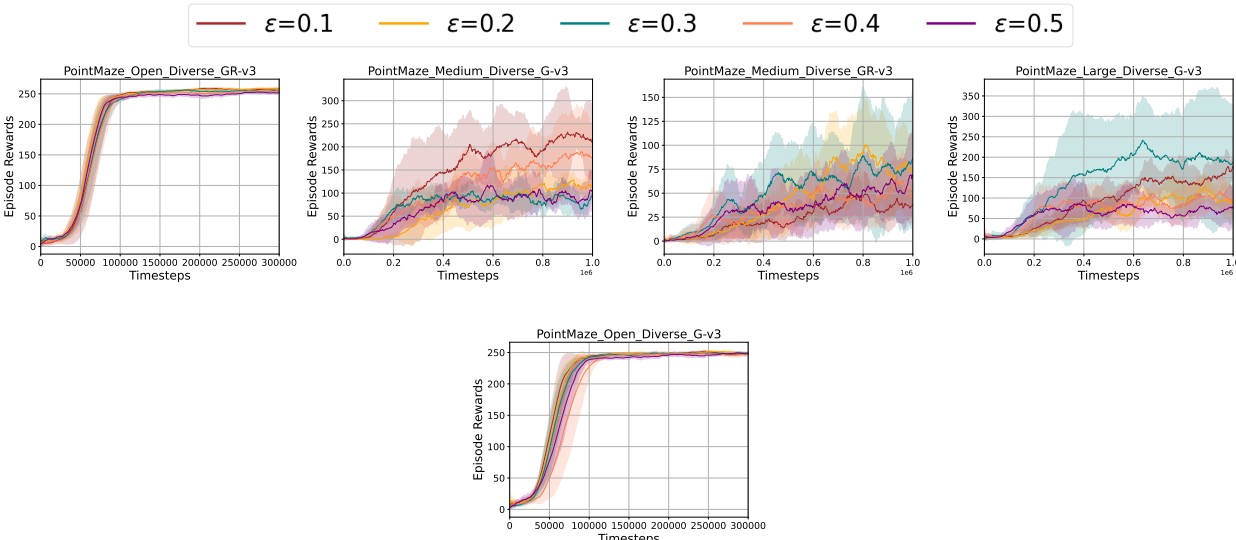

Figure 11: Results show the ablation study on all 5 PointMaze multi-goal sparse reward tasks. The parameter $\epsilon$ controls the replay frequency to balance exploration vs exploitation. The plots are the learning curves and show the episodic rewards along the y-axis evaluated through the current policy with different $\epsilon$. The reported results are across five different seeds.

