# OpenReview forum: "Match or Replay: Self Imitating Proximal Policy"
_TMLR — Rejected by TMLR_

### Review · Reviewer_pdG2 · 2025-03-07

**Summary Of Contributions:**

The paper presents a novel approach, Self-Imitating Proximal Policy (SIPP) algorithm, which enhances exploration and sample efficiency in reinforcement learning across dense and sparse reward settings. Firstly, it introduces a unique self-imitating on-policy algorithm that leverages past successful state-action transitions to bootstrap policy learning. It proposes the Match strategy, utilizing optimal transport methods to prioritize transitions that align with the most rewarding past trajectories in dense reward environments, and the Replay strategy, which maintains an imitation buffer to effectively replay successful trajectories in sparse reward scenarios. These strategies tackle the temporal credit assignment problem and mitigate forgetting behaviors and policy divergence. Additionally, the authors provide extensive empirical validation across various complex environments, demonstrating significant improvements in learning efficiency and performance compared to existing state-of-the-art methods, thereby highlighting the effectiveness and adaptability of the proposed approach.

**Audience:**

Yes

**Claims And Evidence:**

No

**Requested Changes:**

The author claims in the conclusion “through extensive experimentation, we demonstrated that bootstrapping policy learning with past rewarding experiences effectively mitigates forgetting behavior and reduces policy divergence.” Probably, the authors’s conclusion is drawn from the following sentences: “The Replay strategy results in even superior performance, Figure 4, for partially observable environments due to its inherent design capability to adapt to partial observability. The results show that PPO agents encounter some success but fail to learn due to forgetting behaviors. However, the proposed Replay strategy stores those behaviors and keeps replaying them. This addresses the forgetting behaviors, and the agent eventually learns to mimic those successful behaviors.” I understand the logic of the last part. However, it is not clear which part of Figure 4 the author is referring to and why the authors can say that PPO fails to learn due to forgetting behaviors. I do not think any clue is provided. To support this claim, it is necessary to provide empirical evidence.

Though this paper tries to address the exploration challenge, the exploration itself relies solely on the randomness of the current policy. Therefore, for the success of the proposed framework, the task needs to be relatively easy so that the randomness of the behavior policy itself is sufficient to find a positive reward. Otherwise, both Match and Replay strategies barely help to improve the performance of the underlying RL agent. Therefore, in terms of the exploration ability, I don’t see a logical reasoning that supports the claim that the proposed approaches address the exploration challenge. What are the advantages over RND (Burda et al., 2019), ICM (Pathak et al., 2017), and many other exploration strategies? It is unclear what class of tasks the proposed approach is advantageous over the existing techniques. As mentioned above, I suspect that the proposed approach may not work for tasks where exploration is really necessary. Comparisons with other exploration strategies such as RND and ICM on environments requiring exploration (environments used in the papers of these approaches).

The results indeed indicate that the sample efficiency is improved by improving the exploitation ability. Then, the question arises as to whether the proposed approach is advantageous over prioritized experience replay. The proposed approaches are similar to PER in that experiences are selected non-uniform randomly. A comparison with Prioritized Experience Replay both on sparse and dense reward situations would indeed enhance the understanding of the proposed approaches.

**Strengths And Weaknesses:**

Strengths:

Novel Approach: The introduction of the Self-Imitating Proximal Policy (SIPP) algorithm offers a novel strategy for enhancing exploration and sample efficiency in reinforcement learning

Empirical Validation: The extensive experimental results across a range of environments, including MuJoCo, PointMaze, and Animal-AI Olympics, demonstrate improvements in performance over baselines.

Weaknesses:

The focus on self-imitating successful trajectories may limit exploration of novel strategies or unvisited states, potentially leading to suboptimal learning.

Comparison with Other Exploration Techniques: While the paper highlights improvements over existing methods, it lacks a comparative analysis with techniques like PER, RND, and ICM.

Limited Discussion on the target task class: The paper does not extensively discuss potential applications or target class of tasks that the proposed approach is advantageous.

---

> ### Author Response · Authors · 2025-03-17
> **Response to Reviewer pdG2**
>
> We sincerely thank the reviewer for their detailed critique and valuable suggestions. Below, we comprehensively address each weakness and requested change, outlining specific revisions to the manuscript to resolve the concerns raised.
>
>
> **Requested Changes 1: Forgetting Behavior**
>
> - We appreciate the reviewer’s call for clarity regarding forgetting behavior. In the original manuscript, we intended this term to describe how PPO occasionally achieves successful behaviors in Animal-AI tasks but diverges from those behaviors due to its on-policy updates, resulting in performance degradation. Upon reflection, we agree that forgetting behavior may lack precision. We have therefore replaced it with policy instability (divergence), a term more consistent with reinforcement learning (RL) terminology, to describe PPO’s tendency to lose previously learned effective behaviors in sparse reward or partially observable settings. The Replay strategy mitigates policy instability (divergence) by preserving successful behaviors in a buffer and replaying them during training, enabling the agent to reinforce and mimic these behaviors over time. It can be seen in Figures 3  and 4 that the proposed strategy performs better as it continuously reinforces past success behaviors, resulting in better and more stable performance. In the revised manuscript, we have updated all instances of forgetting behavior to policy instability (divergence) and provided a clear definition.
>
> **Requested Changes 2, Weakness 1, 2, and 3: Target Task Class and Exploration**
>
> - While self-imitation in SIPP emphasizes exploiting past successful trajectories, it does not preclude exploration. We achieve a balance between exploration and exploitation via the **Imitation-Exploration Trade-off coefficient ($\epsilon$)**, which controls the proportion of training focused on imitating past successes versus exploring new actions based on the current policy’s randomness. Our ablation study (Figures 5 and 6) demonstrates this flexibility.
>
> - We understand the reviewer's concern regarding the target task class. However, the proposed approach can be extended to any task class, as highlighted in the main text that self-imitation learning has been widely used for many tasks, including robotics(Luo et al., 2023) and LLM(Xiao et al., 2024). We have also demonstrated that SIPP excels in Dense, spare, and binary reward settings, including POMDP.
>
> - The reviewer correctly points out that SIPP relies on the current policy’s randomness for exploration, which might be insufficient for extremely challenging tasks. However, SIPP is designed to complement existing exploration strategies rather than replace them. Once success is achieved—whether through random exploration or an integrated method like RND—SIPP ensures that this success is effectively capitalized upon. Integrating SIPP with any exploration technique can efficiently exploit past successful behaviors. We hypothesize that combining SIPP with RND or ICM techniques could lead to superior performance, leveraging the synergy between exploitation and exploration.
> Moreover, previous research has demonstrated the effectiveness of using demonstrations to enhance learning in difficult exploration scenarios (Salimans et al., 2018; Hester et al., 2017). Our approach can be viewed as a pseudo-demonstration method, relying on self-encountered successes. When coupled with methods such as RND, it holds promise for performance enhancement. However, due to computational constraints (we have increased the number of seeds for MuJoCo and PointMaze, added results on PER, and extended SIL to Animal-AI), we were unable to conduct the recommended extensive experiments. We have highlighted the above insights regarding the integration of SIPP with exploration strategy in limitation and future work direction under the conclusion section of the revised draft.
>
>
> **Requested changes 3: Comparison with PER**
>
> - Traditionally, PER is used in off-policy methods (e.g., DQN), prioritizing experiences based on TD error. For PPO, we implemented a variant where transitions in the rollout buffer are prioritized by TD error and sampled with probabilities matching SIPP’s Match strategy. We tested this PER variant on MuJoCo (dense rewards) and PointMaze (sparse rewards). The PPO+PER didn't result in any performance improvement. On the contrary, it performed inferior to vanilla PPO in both settings. We attribute this to the fact that in policy gradient methods like PPO, TD errors are noisier due to multiple updates per rollout, amplifying bias in prioritization and degrading performance. In contrast, SIPP’s Match strategy prioritizes based on similarity to successful trajectories (state distribution), avoiding reliance on noisy TD errors and reducing bias. We have included the results of PER+PPO in the revised draft.

---

> > ### Author Response · Authors · 2025-03-17
> > **Related citations**
> >
> > [1] Shan Luo and Lambert Schomaker. Reinforcement learning in robotic motion planning by combined experience-based planning and self-imitation learning. Robotics Auton. Syst., 170:104545, 2023. URL https://api.semanticscholar.org/CorpusID:259137688.
> >
> > [2] Teng Xiao, Mingxiao Li, Yige Yuan, Huaisheng Zhu, Chao Cui, and V.G. Honavar. How to leverage demonstration data in alignment for large language model? a self-imitation learning perspective. In Conference
> > on Empirical Methods in Natural Language Processing, 2024. URL https://api.semanticscholar.org/
> > CorpusID:273345418.
> >
> > [3] Salimans, Tim, and Richard Chen. "Learning montezuma's revenge from a single demonstration." arXiv preprint arXiv:1812.03381 (2018).
> >
> > [4]Todd Hester, Matej Vecerik, Olivier Pietquin, Marc Lanctot, Tom Schaul, Bilal Piot, Dan Horgan,
> > John Quan, Andrew Sendonaris, Gabriel Dulac-Arnold, et al. Deep q-learning from demonstrations.
> > arXiv preprint arXiv:1704.03732, 2017.

---

> > ### Comment · Reviewer_pdG2 · 2025-03-19
> >
> > Thank you for your response.
> >
> > For the second point, though I understand the logic of the author, I have to say that the current empirical results do not really support the claim, i.e., the proposed approach enhances exploration. Moreover, without any empirical results, we can not judge whether the proposed approach really helps performance when employed with other exploration strategies.
> >
> > For the third point, thank you for the additional experiments. An additional question is why an on-policy type algorithm is necessary in this problem setup. It seems to me that there is no clear motivation. Please clarify this point.

---

> > > ### Author Response · Authors · 2025-03-31
> > >
> > > **Integrating RND+SIPP:** We appreciate the reviewer's feedback. Based on your suggestion, we have integrated RND with SIPP, which can be found in Appendix A of the revised draft. We test the performance of RND+SIPP on the three hard exploration tasks, the same as RND (Burda et al., 2018).
> > >
> > > In the RND+SIPP framework, the agent’s total reward at each timestep is the sum of the extrinsic reward from the environment and the intrinsic reward from RND. The policy is updated via PPO with the SIPP Replay strategy, where the imitation buffer stores trajectories based solely on their cumulative extrinsic rewards. This ensures that SIPP reinforces behaviors leading to tangible environmental success while RND independently drives the exploration of novel regions, mitigating potential conflicts between exploitation and exploration objectives.
> > >
> > > **Table 1: Performance Comparison on Hard Exploration Tasks.**
> > >
> > > | Task       | RND    | RND+SIPP |
> > > |-------------|---------|----------------|
> > > | Gravitar   | 3906 | 4363     |
> > > | Venture   | 1859 | 1813     |
> > > | Solaris    | 3282 | 3589     |
> > >
> > > Integrating SIPP with RND demonstrates that combining self-imitation learning with intrinsic motivation provides a dual benefit. On the one hand, SIPP ensures that the agent leverages its past successes to stabilize policy updates. On the other hand, RND continually drives the agent to explore unvisited or less familiar regions of the state space. The Imitation- Exploration Trade-off coefficient controls the trade-off between these components, allowing for task-specific tuning.
> > >
> > > **Motivation for on-policy:** Most prior research on self-imitation learning (SIL) has centered on off-policy algorithms. Self-imitation, which involves learning from an agent’s own successful past behaviors, aligns naturally with off-policy approaches to leverage historical data. As a result, integrating self-imitation into off-policy frameworks has been relatively straightforward. In contrast, integrating self-imitation with on-policy algorithms is more challenging and has received less attention, leaving a gap in understanding. Our work addresses this directly by developing the Self-Imitating Proximal Policy Optimization (SIPP) framework, which integrates self-imitation into the on-policy algorithm PPO. A discussion on this can be found in the Introduction section.
> > >
> > > [1] Burda, Yuri, et al. "Exploration by random network distillation." arXiv preprint arXiv:1810.12894 (2018).

---

### Review · Reviewer_Gajt · 2025-03-07

**Summary Of Contributions:**

The paper proposes algorithms based on self-imitation learning, where an agent utilizes its own past behavior to tackle the challenges of temporal credit assignment and exploration in on-policy RL. To this end, it outlines two methods: a) Match, designed for dense reward settings, which calculates the Wasserstein distance between a trajectory and previous successful trajectories, unweighting those that are more similar to past high-performing ones; and b) Replay, tailored for sparse reward scenarios, which reuses past high-return experiences for policy updates, blending them with updates from newly gathered data. Experiments in both dense and sparse reward environments, benchmarked against earlier self-imitation learning studies, demonstrate that the proposed methods are competitive.

**Audience:**

Yes

**Claims And Evidence:**

No

**Requested Changes:**

Critical to securing recommendation:

1. Please clearly delineate your contributions in relation to prior work on self-imitation learning. For instance, claims like _“Oh et al. address exploration and self-imitation for off-policy algorithms, while we focus on on-policy algorithms”_ are imprecise, as both approaches use PPO as the underlying RL algorithm. Additionally, Oh et al.’s method is compatible with other on-policy algorithms like A2C.

2. Please provide a stronger motivation for employing self-imitation in dense reward environments. What specific issue does it address to enhance sample efficiency? If exploration is the primary motivation, consider dense reward environments where exploration poses a challenge, as MuJoCo may not be an ideal testbed for evaluating improvements in exploration skills.

3. Please include background on key components such as PPO, Optimal Transport, the advantage function, and value networks—all of which are used in the algorithm box without much explanation.

4. Please discuss any computational overheads associated with OT, if applicable, and elaborate on how it scales with increasing state dimensionality.

5. Much of Section 4.2 overlaps with prior self-imitation work. Please explicitly highlight what is new in this section to distinguish it from existing approaches.

6. In Section 5.2, SVPG is not the algorithm proposed by Gangwani et al.; rather, they use it in their work. Please correct this. Also, the statement _“Gangwani et al. (2018) use KL-divergence as a regularizer to minimize divergence…”_ appears inaccurate. Please verify.

7. It appears that, in Figure 3, the value of $\epsilon$ was selectively chosen for each environment to yield the best result (suggested by Figure 5). This is slightly misleading and amounts to peeking into the test set to adjust hyperparameters to improve test performance.

8. The baselines could be adapted to work in partially observable environments, similar to how SIPP operates (e.g., by treating the concatenation of the past four observation frames as the "state"). Please include these baselines in your evaluation for POMDP experiments.

9. Please acknowledge the limitations of self-imitation learning, such as the risk of getting trapped in local minima in environments with deceptive rewards, to provide a balanced perspective on the approach.

**Strengths And Weaknesses:**

Strengths:
1. Addressing temporal credit assignment and exploration challenges, particularly in sparse reward settings, is a significant and valuable endeavor, as these are critical hurdles for applying RL effectively in real-world scenarios
2. The experiments span a diverse range of environments, covering both dense and sparse reward structures, as well as fully observed and partially observed setups

Weakness:
1. The paper fails to distinctly highlight its contributions relative to prior work in self-imitation learning. The approach of storing successful trajectories in a replay buffer and leveraging them for policy updates has thoroughly explored in several cited studies
2. The paper lacks self-sufficiency, omitting key details about the base methods underpinning the proposed approach
3. The paper lacks a clear explanation of how its self-imitation approach overcomes the shortcomings of prior efforts, and the empirical evidence provided does not effectively support specific, well-defined claims

---

> ### Author Response · Authors · 2025-03-17
> **Response to Reviewer Gajt**
>
> We thank the reviewer for their critical feedback and acknowledge the need for greater precision in articulating our contributions relative to prior self-imitation learning (SIL) methods. Below, we address each concern and outline specific revisions to the manuscript to resolve them.
>
> **Requested changes 1 and weakness 1: off-policy nature of SIL**
>
> We agree with the reviewer that SIL uses PPO and A2C on-policy algorithms. However, they use a modified off-policy actor-critic loss function for A2C. Further, (Oh et al. 2018) presented their approach as an off-policy algorithm. They made this distinction clear in their work. They also suggest that their approach doesn't have a theoretical grounding for on-policy algorithms like PPO, but they extended SIL to PPO. This is primarily because they store past experiences (maintain a replay buffer) and prioritize state-action pairs where returns exceed value estimates. This introduces an off-policy flavor, even when applied to PPO, potentially destabilizing on-policy learning due to bias from off-policy data. Specifically, we don't maintain any replay buffer to better align with on-policy RL. Also, we don't restrict our algorithm to dense or delayed rewards like prior work. These distinctions have been highlighted in the Introduction section of the revised manuscript.
>
> **Requested Changes 2: motivation for employing self-imitation in dense rewards**
>
> We acknowledge the need for stronger motivation to use self-imitation in dense reward environments like MuJoCo. Even in dense reward settings, RL agents can struggle with sample efficiency due to high-dimensional state and action spaces, leading to slow convergence or suboptimal policies. SIPP’s Match strategy addresses this by guiding exploration toward high-value regions using optimal transport to align current trajectories with past successful ones, enhancing learning speed and performance. Empirical results (Figure 1) show SIPP-Match outperforming other baselines' final performance and convergence rate, demonstrating improved sample efficiency. We have revised Section 1 to clarify that self-imitation refines exploration and reduces policy divergence in dense reward settings.
>
> **Requested changes 3 and weakness 2: Background on the key component**
>
> In the appendix, we have added a discussion on the OT, PPO, value network, and advantage function.
>
> **Requested Changes 4: Computational complexity**
>
> In SIPP-Match, Optimal Transport (OT) introduces manageable overhead via the Sinkhorn algorithm, which computes the Wasserstein distance between the current rollout’s state visitation and the best past trajectory stored in the imitation buffer. With only one trajectory stored, the complexity is $\mathcal{O}(n^2logn),$ where $n$ is trajectory length. The OT cost function (e.g., Euclidean distance) scales linearly with state dimension, but the Sinkhorn algorithm’s complexity depends primarily on $n$, not dimensionality. Further, The OT computation occurs once per policy update, with results reused for sampling, ensuring efficiency.
>
> **Requested Changes 5 and weakness 3: Distinction of replay from existing approach**
>
> While the Replay strategy (Section 4.2) shares conceptual similarities with prior SIL methods, SIPP-Replay introduces novel elements for on-policy RL in sparse and binary reward settings:
>
> - Trajectory-Level Replay: Unlike Oh et al. (2018), which prioritizes individual state-action pairs, SIPP-Replay uses entire successful trajectories, addressing temporal credit assignment in sparse reward tasks.
>
> - On-Policy Integration: It samples trajectories from the rollout or imitation buffer (controlled by IET), avoiding off-policy corrections and preserving PPO’s stability. Figure 2 shows that past approaches designed for dense or delayed rewards do not perform similarly for spare and binary reward tasks such as PointMaze and AnimalAI (Partially observable three-dimensional environment).
>
> - POMDP Compatibility: As shown through direct applicability to Animal-AI Olympics tasks, it adapts to partial observability without requiring state visitation distributions.
>
> These points highlight the differentiation of Replay while highlighting its border applicability.
>
> **Requested Changes 6: Inaccuracy  in baseline text**
>
> We apologize for the errors. Gangwani et al. (2018) proposed the Stein Variational Policy Gradient (SVPG), using Stein variational gradient descent to minimize divergence between the current policy’s visitation and past high-return trajectories, not KL-divergence. We have made the correction in the revised draft.
>
> **Requested Changes 7: selection of $\epsilon$**
>
> We clarify that the value of IET coefficient $\epsilon=0.3$ was fixed across all tasks except for PointMaze\_Medium\_Diverse\_G, where we use $\epsilon=0.1$. We have also highlighted in the ablation study of the main text that performance is most consistent for the IET coefficient $\epsilon=0.2,0.3$ across domains.

---

> > ### Author Response · Authors · 2025-03-17
> > **Response to Reviewer Gajt**
> >
> > **Requested Changes 8:Adaption of baseline for AnimalAI**
> >
> > While baselines like SIL (Oh et al., 2018) and SVPG (Gangwani et al., 2018) could theoretically use observation histories, their reliance on state visitation distributions complicates adaptation to POMDPs, requiring significant modifications.
> > However, based on your feedback, we modified SIL to adapt to the task requirement of the AnimalAI environment. The results have been incorporated in the revised draft. The SIL performs poorly than vanilla PPO. The performance drop is due to the nature of the SIL algorithm, which depends on a dense reward structure. The performance drop is justified since AnimalAI is a binary reward environment with partial observability. This further highlights the advantage of our method, which focuses on trajectory-level imitation rather than on states and also supports the need for different strategies for dense and sparse reward environments as proposed in this work.
> >
> > **Requested Changes 9: Limitation of SIPP**
> >
> > We agree that self-imitation learning risks local optima in deceptive reward settings, where early successes may hinder global exploration, and have included the limitation of our work in the conclusion section. However, we would like to highlight that the risk of getting trapped in local minima can be addressed by integrating our approach with an exploration-based strategy such as RND. As RND performs better in hard exploration tasks, building self-imitation on top of that can perform even better as exploration and exploitation complement each other.

---

### Review · Reviewer_UgaX · 2025-03-08

**Summary Of Contributions:**

This paper presents Self-Imitating Proximal Policy (SIPP), a reinforcement learning method combining self-imitation with policy optimization. SIPP features two strategies:

Match Strategy (Dense Rewards): Uses optimal transport (Sinkhorn distance) to highlight valuable past trajectories.

Replay Strategy (Sparse Rewards): Replays successful past experiences to improve credit assignment.

Tests across MuJoCo, PointMaze, and Animal-AI Olympics demonstrate SIPP's superior learning speed and performance, highlighting the effectiveness of reusing successful experiences.

**Audience:**

No

**Broader Impact Concerns:**

The paper does not explicitly address broader ethical implications. However, given that reinforcement learning algorithms are increasingly deployed in real-world applications (e.g., autonomous driving, robotics), it would benefit from a brief discussion on potential ethical concerns or risks associated with the misuse or unintended consequences of deploying advanced RL methods.

**Claims And Evidence:**

No

**Requested Changes:**

**Critical Changes Required**

- Polish the statemnt and clarify some vague presentations.
- Provide clear theoretical reasoning or more detailed explanations for how exactly self-imitation aids temporal credit assignment.
- Explain more explicitly why certain baselines (especially in Animal-AI Olympics) were excluded due to partial observability. If possible, consider including at least one additional baseline adapted for partial observability or justify more clearly why they cannot be fairly adapted.

**Important but Non-critical** (would significantly strengthen the paper):

- Add guidelines or further insights for tuning the hyperparameter ϵ.
- Discuss briefly ethical and broader impacts to align with current best practices in RL research.

**Strengths And Weaknesses:**

# Strength:

- The paper proposed a self-imitation learning, particularly emphasizing an on-policy context, which is relatively less explored compared to off-policy settings.
- Clear distinction and targeted strategies (Match and Replay) for different reward structures (dense vs. sparse) significantly enhance method practicality and application potential.
- The combination of Optimal Transport methods with self-imitation is novel and provides theoretical elegance as well as practical efficiency.
- The extensive experiments across diverse domains effectively demonstrate robustness and broad applicability.

# Weakness:
## Writing

The writing of the paper is not very clear. Some statement are vague, invalid or contains logical issues.



Title:

I don't understand why the method is called "Self Imitating Proximal Policy". Is the proposed method a policy?



Abstract:

- "This inefficiency comes from unsystematic exploration, where agents fail to effectively exploit past successful experiences ". I don't understand why exploration is related to exploiting past successful experiences.
- what do you mean by "bootstrapping policy learning "?
- what do you mean sucessful for " state-action transitions"?

Preliminaries:

- Is $S_0$ a set or fixed state? It is more typical to use a initial distribution.
- Please use "$\mathbb{R}$" for the set of real numbers.



Method:

- It is not clear what do you refer to "experience". "The proposed approach connects with self-imitation learning by prioritizing experiences similar to the past most rewarding episodic rollout"
- Algorithm 1, please make it more clear how to update $\mathcal{B}_I$, how to collect rollout..
- Algorithm 2, what do you mean by "Replay a trajectory from...". There seems to be issues in Line 8.
- (minor) I bit suspect that whether the method could be considerred as a self-imitation learning method. It looks more like a priorized experinece replay method.

Experirment:

- "SIL lacks a theoretical connection with on-policy algorithms", does your method have this  connection?



## Technique:



1. "Unlike prior works (Oh et al., 2018;Li et al., 2023; Tang, 2020), which mostly addresses exploration and self-imitation for off-policy algorithms, we focus on on-policy algorithms. " To me, this does not hold, because the algorithm still need off-policy data. Besides, this could increase the complexity.
2. For the replay method, it will use off-policy data from previous experience. This could result in bias.
3. How to unify the two methods in the same framework?
4. The claim about reduced forgetting is intuitively appealing but lacks rigorous empirical/theoretical justification.
5. Reducing Forgetting is not very clear to me: Expand on how the method concretely reduces forgetting. Does frequent exposure to similar successful experiences inherently stabilize the policy? If so, provide more explicit reasoning or experiments supporting this assertion.





## Experiment:

- Running with only 5 seeds is not enough.
- The baselines are not sufficient. The paper only compare to algorithms proposed in 6 years ago.
- In Animal-AI Olympics environment, the method is only compared to PPO.

---

> ### Author Response · Authors · 2025-03-17
> **Response to Reviewer UgaX**
>
> We thank the reviewer for the detailed review comments. Your feedback has helped enhance the clarity of the work. Below, we address all the concerns raised. We hope this addresses all your comments.
>
> **Writing**
>
> We have made appropriate changes in the revised draft to address all the writing concerns.
>
> - Title: The title may suggest the method is a policy, though we had explicitly mentioned that the proposed approach is Algo in the original draft. We have revised it to "Self-Imitating Proximal Policy Optimization" for clarity.
>
> - Exploration vs. Exploitation: Exploration in reinforcement learning (RL) means discovering new strategies while exploiting past successful experiences, which means using known high-reward actions. Inefficiency occurs when agents cannot balance these, and our method addresses this by guiding exploration with past successes. We have made this distinction more explicit.
>
> - Bootstrapping Policy Learning: This refers to accelerating learning using the agent’s past successful experiences, akin to resampling in statistics. We have clarified this in the revised draft.
>
> - Successful State-Action Transitions: These transitions (state, action, next state) lead to high rewards. We have changed the notation with a clear definition.
>
> - $S_0$: This is the initial state distribution, not a fixed state. We have clarified this notation.
>
> - Experience: This term refers to state-action transitions. Our Match strategy prioritizes transitions similar to the most rewarding past trajectory, measured via optimal transport, aligning with self-imitation learning.
>
> - Algorithm 1: $\mathcal{B_I}$ is updated by storing the trajectory with the highest return seen so far. Rollouts are collected using the current policy $\pi$. We have clarified this.
>
> - Algorithm 2: "Replay a trajectory from $\mathcal{B_I}$
>  "means sampling a past successful trajectory and adding its transitions to the rollout buffer $\mathcal{D}$. Line 8 will be corrected to specify sampling transitions for optimization.
>
> - Self-Imitation vs. Prioritized Experience Replay: Unlike prioritized experience replay, which focuses on individual transitions based on TD error, our method emphasizes trajectory-level imitation and distribution matching. To clarify the distinction, we added results corresponding to the PER strategy in the revised draft.
>
> - SIL Comparison: Unlike SIL (Oh et al., 2018), which lacks a strong theoretical link to on-policy methods, our approach integrates fully with PPO’s on-policy framework, preserving its theoretical guarantees. The referenced statement was taken from (oh et al., 2018). This is primarily because they maintain a replay buffer to store past experiences and prioritize state-action pairs where returns exceed value estimates. This introduces an off-policy flavor, even when applied to PPO, potentially
> destabilizing on-policy learning due to bias from off-policy data.

---

> > ### Author Response · Authors · 2025-03-17
> > **Response to Reviewer UgaX**
> >
> > **Technique**
> >
> > - On-Policy vs. Off-Policy Data: The proposed approach operates within an on-policy framework by leveraging past experiences in a controlled manner that is distinct from traditional off-policy approaches. Specifically, SIPP modifies the on-policy rollout buffer’s sampling strategy (Match) or selectively replays successful trajectories (Replay). Unlike off-policy methods that depend on large replay buffers, complex importance sampling corrections, or even modified loss function (SIL) to integrate self-imitation learning, SIPP avoids such mechanisms by using data generated under the current policy or closely aligned successful trajectories.
> >
> > - Bias in replay method: The replay method replays all successful trajectories, treating them as if they reflect the current policy’s behavior. This approach resembles importance sampling but is regulated by a hyperparameter, $\epsilon$, which limits the proportion of replayed data. This controlled use of $\epsilon$ prevents over-reliance on outdated trajectories, reducing bias compared to unrestricted off-policy replay. Further empirical results suggest that this strategy enhances learning rather than introducing bias.
> >
> > - Unifying the Match and Replay Methods: Both methods modify how data is sampled or generated for PPO updates, adapting to different reward structures—Match prioritizes state distributions similar to the most rewarding trajectories for dense-reward settings, while Replay leverages successful past trajectories for sparse-reward scenarios. This adaptability is the core of SIPP, as it tailors the data selection process using past self-encountered trajectories to enhance learning efficiency within the on-policy PPO framework.
> >
> > - Reducing Forgetting: SIPP reduces catastrophic forgetting by prioritizing or replaying successful experiences, reinforcing high-reward behaviors during policy updates. In standard on-policy methods like PPO, frequent updates with new data can overwrite previously learned behaviors, especially in sparse-reward tasks. SIPP counteracts this by ensuring the policy retains exposure to valuable past successes. Upon reflection, we agree that
> > forgetting behavior may lack precision. We have therefore replaced it with policy instability, a term more
> > consistent with reinforcement learning (RL) terminology, to describe PPO’s tendency to lose previously
> > learned effective behaviors in sparse reward or partially observable settings
> >
> > **Experiment**
> >
> > - Seed: We raised seeds from 5 to 7 for the Mujoco and PointMaze environment.
> >
> > - Baselines: Our primary focus is on self-imitation learning (SIL), where the baselines from Oh et al. (2018) and Gangwani et al. (2018) remain the most relevant and widely cited for on-policy RL. Recent advancements in SIL (e.g., Luo et al., 2021; Shi et al., 2023) tend to be domain-specific rather than offering general improvements to the SIL framework, limiting their applicability to our study.
> >
> > - We have also extended SIL for the Animal-AI Olympic.
> >
> > **Requested Changes 1:
> > Polishing Statements and Clarifying Vague Presentations:*** We have addressed all the writing suggestions in the revised draft, as highlighted above.
> >
> > **Requested Changes 2: Temporal credit assignment** In RL, temporal credit assignment refers to the challenge of determining which actions in a sequence are responsible for a reward, particularly when rewards are delayed or sparse. For example, in a game where a reward is given only at the end, the agent must learn which earlier actions contributed to that outcome. Self-imitation involves the agent learning from its own past successful experiences by prioritizing or replaying them during training. This process strengthens the agent’s ability to connect actions to their long-term consequences through repeated exposure to success and reducing noise from exploration. This approach builds on prior work, such as experience replay in Deep Q-Networks (Mnih et al., 2015), which showed that revisiting key experiences improves learning efficiency and early self-imitation ideas (Lin, 1992), highlighting the benefit of learning from past successes.
> >
> > **Requested changes 3: Limited Comparison in Animal-AI:** As noted in our manuscript, existing SIL baselines like SIL and SVPG rely on state visitation distributions, which are not directly applicable to partially observable environments (POMDPs) like Animal-AI Olympics. Adapting these methods would require significant changes, potentially altering their core mechanisms. Since SIPP is, to our knowledge, the first SIL method evaluated in a POMDP setting, comparing it to vanilla PPO—a widely used RL algorithm—serves as a meaningful baseline to highlight the benefits of self-imitation in partially observable environments. However, we have extended SIL (we stack multiple frames together) to Animal-AI environments, and it performed inferior to PPO as SIL, by nature, depends on dense or sparse reward structures contrary to binary rewards in Animal-AI.

---

> ### Author Response · Authors · 2025-03-17
> **Response to Reviewer UgaX**
>
> **Suggested changes 1: Guidelines for tuning $\epsilon$** We have added guidelines ( Appendix B) to tune $\epsilon$ and added further insights in the revised manuscript.
>
> **Suggested changes 2: Ethical and broader impact concern** We have added these in Appendix C.
>
>
> [1] Junhyuk Oh, Yijie Guo, Satinder Singh, and Honglak Lee. Self-imitation learning. In International conference
> on machine learning, pp. 3878–3887. PMLR, 2018.
>
> [2] Tanmay Gangwani, Qiang Liu, and Jian Peng. Learning self-imitating diverse policies. arXiv preprint
> arXiv:1805.10309, 2018.
>
> [3] Sha Luo, Hamidreza Kasaei, and Lambert Schomaker. Self-imitation learning by planning. In 2021 IEEE
> International Conference on Robotics and Automation (ICRA), pp. 4823–4829. IEEE, 2021.
>
> [4] Zijing Shi, Yunqiu Xu, Meng Fang, and Ling Chen. Self-imitation learning for action generation in text-based
> games. In Conference of the European Chapter of the Association for Computational Linguistics, 2023.
> URL https://api.semanticscholar.org/CorpusID:258378233.
>
> [5]  Lin, Long-Ji. "Self-improving reactive agents based on reinforcement learning, planning and teaching." Machine learning 8 (1992): 293-321.
>
>  [6] Mnih, Volodymyr, et al. "Human-level control through deep reinforcement learning." nature 518.7540 (2015): 529-533.

---

> > ### Comment · Reviewer_UgaX · 2025-05-13
> > **Furthere comments**
> >
> > Thanks to the authors for their detailed feedback. I still have several concerns about the statements made in the paper.
> >
> > > SIL Comparison: Unlike SIL (Oh et al., 2018), which lacks a strong theoretical link to on-policy methods, our approach integrates fully with PPO’s on-policy framework, preserving its theoretical guarantees. The referenced statement was taken from (oh et al., 2018). This is primarily because they maintain a replay buffer to store past experiences and prioritize state-action pairs where returns exceed value estimates. This introduces an off-policy flavor, even when applied to PPO, potentially destabilizing on-policy learning due to bias from off-policy data.
> > >
> > > Bias in replay method: The replay method replays all successful trajectories, treating them as if they reflect the current policy’s behavior. This approach resembles importance sampling but is regulated by a hyperparameter, ϵ, which limits the proportion of replayed data. This controlled use of ϵ prevents over-reliance on outdated trajectories, reducing bias compared to unrestricted off-policy replay. Further empirical results suggest that this strategy enhances learning rather than introducing bias.
> >
> > I agree that your method can reduce bias by tuning the hyperparameter $\epsilon$. However, I believe the paper should not claim that "SIPP avoids the complexity and potential bias associated with off-policy data."
> >
> > Since the method uses off-policy data, it inherently involves some level of bias. Even with importance sampling, this bias is not entirely eliminated. I recommend referring to the GePPO paper for further insights into this issue:
> >
> > [1] Queeney J, Paschalidis Y, Cassandras C G. *Generalized proximal policy optimization with sample reuse*. NeurIPS, 2021.
> >
> > ------
> >
> > > **Requested Changes 2: Temporal credit assignment** In RL, temporal credit assignment refers to the challenge of determining which actions in a sequence are responsible for a reward, particularly when rewards are delayed or sparse. For example, in a game where a reward is given only at the end, the agent must learn which earlier actions contributed to that outcome. Self-imitation involves the agent learning from its own past successful experiences by prioritizing or replaying them during training. This process strengthens the agent’s ability to connect actions to their long-term consequences through repeated exposure to success and reducing noise from exploration. This approach builds on prior work, such as experience replay in Deep Q-Networks (Mnih et al., 2015), which showed that revisiting key experiences improves learning efficiency and early self-imitation ideas (Lin, 1992), highlighting the benefit of learning from past successes.
> >
> > I still don’t see how changing data priority relates directly to credit assignment. As you mentioned, "temporal credit assignment refers to the challenge of determining which actions in a sequence are responsible for a reward." Does your method explicitly decide how to assign credit or distribute reward to specific actions over time? If not, I suggest removing this claim from the paper, as it may be misleading.
> >
> > ------
> >
> > > As noted in our manuscript, existing SIL baselines like SIL and SVPG rely on state visitation distributions, which are not directly applicable to partially observable environments (POMDPs) like Animal-AI Olympics. Adapting these methods would require significant changes, potentially altering their core mechanisms.
> >
> > This point is still unclear. PPO is also based on state visitation distributions, and since your method builds on PPO, I assume the same limitation applies. Could you clarify what makes SIL and SVPG less applicable here compared to your method?

---

> > > ### Author Response · Authors · 2025-05-15
> > >
> > > We sincerely thank the reviewer for their continued thoughtful engagement. Below, we address each of your remaining concerns in detail and have updated the manuscript to reflect these clarifications.
> > >
> > > **1. On the claim that SIPP avoids off-policy bias:**
> > > We agree that our original phrasing overstated the extent to which SIPP avoids off-policy bias. While our approach integrates replayed trajectories directly into PPO’s on-policy loop and does not require off-policy corrections like importance sampling or target networks, it still involves data reuse, which inherently introduces some level of bias. In response, we have revised the relevant text (Section 4) in the manuscript to state:
> > >
> > >     SIPP reuses successful trajectories in a controlled fashion and integrates them into PPO’s on-policy training loop, without requiring additional target networks or off-policy correction mechanisms such as importance sampling or density ratio estimation. While this introduces some degree of bias due to sample reuse, as discussed in works such as GePPO (Queeney et al., 2021), empirical results indicate that this approach remains stable and effective.
> > >
> > > **2. On the connection to temporal credit assignment:** We appreciate the clarification. In the revised paper, we have removed references to temporal credit assignment to avoid overclaiming. Our intent was to highlight that Replay helps in sparse-reward environments by increasing exposure to high-return trajectories.
> > > This repeated exposure enables the agent to reinforce behavior sequences that led to success and strengthens PPO’s learning signal in long-horizon tasks. We now describe this more accurately as (Section 4.2 and Introduction):
> > >
> > >     Replay improves learning in sparse-reward settings by repeatedly exposing the agent to high-return trajectories, thereby reinforcing long-term action dependencies and supporting more effective learning from delayed rewards.
> > >
> > > **3. On the applicability of SIL and SVPG in partially observable environments:** We thank the reviewer for raising this important point. While PPO, like all policy gradient methods, operates over the policy's trajectory distribution and uses advantage estimates, it does not require explicit modeling of the state visitation distribution or direct comparisons between past and current policy outcomes. This distinction becomes critical in partially observable environments.
> > > - SIL  explicitly stores past trajectories and selectively updates the policy based on whether the trajectory return exceeds a value estimate. This requires comparing off-policy returns to current estimates, which is known to introduce bias and instability, particularly in POMDPs, where observations are partial and trajectories may no longer reflect the current policy’s distribution.
> > >
> > > - SVPG similarly regularizes the policy based on divergence between visitation distributions, which assumes the ability to estimate or match distributions over states—an assumption that breaks down under partial observability..
> > >
> > > In contrast, SIPP avoids these pitfalls by treating successful past trajectories as expert demonstrations and directly integrating them into PPO's update loop. This requires no divergence estimation and no comparisons between return and value predictions. As a result, SIPP remains stable and effective in POMDPs without requiring access to full state information or calibrated value targets.
> > >
> > > To empirically validate this, we adapted the SIL baseline by feeding in observation histories as proxy states in the Animal-AI Olympics environment. Despite this, SIL performed worse than even vanilla PPO. We attribute this to its sensitivity to mismatch between stored and current trajectories—something that SIPP robustly avoids by fully integrating successful rollouts into training without explicit corrections.
> > >
> > > We thank the reviewer again for their insightful comments. The revised manuscript now more accurately reflects the capabilities and limitations of our method, and we believe these changes have substantially strengthened the clarity and rigor of the work.
> > >
> > > If there are any remaining questions or suggestions, we would be happy to address them.

---

### Decision · Action_Editor_ktMx · 2025-05-20

**Recommendation:** Reject

**Comment:**

We received three expert reviews for this paper. The reviewers acknowledged the importance of the problem addressed in the paper and appreciated the empirical validation on a diverse set of environments.    At the same time, the reviewers have raised multiple serious concerns about the claims of the paper. As the AE, I have also read the paper, the reviewers, and the rebuttal by the authors. I agree with most of the comments by the reviewers,  please see the section on “Claims and Evidence” above, where I have summarized these concerns. The paper will also greatly benefit from a clearer articulation of its novelty relative to prior work and more concrete guidance on its applicability.

**Audience:**

Audience: This paper is likely to interest a subset of TMLR’s audience, particularly researchers working on improving the sample efficiency of RL. However, the appeal is limited by the multiple issues, including (i) the paper overstates novelty relative to prior SIL work, and (ii) the claims are ambiguous and often not supported by rigorous analysis or experiments (e.g., on-policy nature, forgetting, credit assignment).

**Claims And Evidence:**

The major claims of the papers are the following:

1.	SIPP is a fully on-policy method: More than one reviewer has questioned this claim during the review. The replay strategy uses an imitation buffer that reuses past trajectories, introducing off-policy components. While the authors tried to address this concern during the rebuttal, the reviewers are still not convinced, see the responses of Reviewer Gajt, Reviewer UgaX.
2.	SIPP improves temporal credit assignment: The method prioritizes or reuses high-return trajectories but does not include any explicit mechanism (e.g., backward credit assignment, reward redistribution) for credit assignment over time. Reviewers found this claim misleading and unsupported, see the responses of Reviewer UgaX, Reviewer Gajt.
3.	SIPP reduces policy forgetting: Reviewer pdG2 and Reviewer UgaX have pointed out that this claim lacks empirical or theoretical evidence (e.g., ablations showing stability over time or recovery from regression). The observed improvements shown in the experiment could stem from exploitation rather than true mitigation of forgetting.
4.	 SIPP enhances exploration: While SIPP improves performance, the mechanism relies on behavioral randomness rather than a clear exploration-exploitation trade-off design. Reviewers noted that the method may fail in hard-exploration tasks unless the environment occasionally delivers rewards through chance. The additional experiments provided as a response to Reviewer Gajt and Reviewer pdG2 do not clearly show the benefits of SIPP’s enhanced exploration.
5.	 SIPP improves sample efficiency and performance in benchmark tasks: Reviewers acknowledge that these claims are sufficiently supported by experiments.